



# An investigation of mesoscale wind direction changes and their consideration in engineering models

Anna von Brandis[1], Gabriele Centurelli[2], Jonas Schmidt[1], Lukas Vollmer[1], Bughsin' Djath[3], and
Martin Dörenkämper[1]

[1]Fraunhofer IWES, Küpkersweg 70, D-26129 Oldenburg, Germany
[2]ForWind - Center for Wind Energy Research, University of Oldenburg, Küpkersweg 70, D-26129 Oldenburg, Germany
[3]Institute of Coastal Systems, Helmholtz-Zentrum Hereon, Max-Planck-Str. 1, D-21502 Geesthacht, Germany

**Correspondence:** Martin Dörenkämper (martin.doerenkaemper@iwes.fraunhofer.de)

**Abstract.** We propose that considering mesoscale wind direction changes in the computation of wind farm cluster wakes could reduce the uncertainty of engineering wake modelling tools. The relevance of mesoscale wind direction changes is investigated using a wind climatology of the German Bight area covering 30 years, derived from the New European Wind Atlas (NEWA). Furthermore, we present a new solution for engineering modelling tools that accounts for the effect of such changes on the

propagation of cluster wakes. Mesoscale wind direction changes are found to exceed $7°$ per $100\,\mathrm{km}$ in $50\,\%$ of all cases and are particularly large in the lower partial load range, which is associated with strong wake formation. Here, the quartiles reach up to $20°$ per $100\,\mathrm{km}$. Especially on a horizontal scale of several tens to a hundred kilometers, wind direction changes are relevant. Both the temporal and spatial scale at which large wind direction changes occur depend on the presence of pressure systems. Furthermore, atmospheric conditions which promote far-reaching wakes were found to align with a strong turning in

$14.6\,\%$ of the cases. In order to capture these mesoscale wind direction changes in engineering model tools, a wake propagation model was implemented into the Fraunhofer IWES wind farm and wake modelling software *flappy*. The propagation model derives streamlines from the horizontal velocity field and forces the single turbine wakes along these streamlines. This model has been qualitatively evaluated by simulating the flow around wind farm clusters in the German Bight with data from the mesoscale atlas of NEWA and comparing the results to Synthetic Aperture Radar (SAR) measurements for selected situations.

The comparison reveals that the flow patterns are in good agreement if the underlying mesoscale data capture the velocity field well. For such cases, the new model provides an improvement compared to the baseline approach of engineering models, which assumes a straight-line propagation of wakes. The streamline and the baseline model have been further compared in terms of their quantitative effect on the energy yield. Simulating two neighbouring wind farm clusters over a time period of 10 years, it is found that there are no significant differences across the models when computing the total energy yield of both clusters.

However, extracting the wake effect of one cluster on the other, the two models show a difference of about $1\,\%$. Even greater differences are commonly observed when comparing single situations. Therefore, we claim that the model has the potential to reduce uncertainty in applications such as site assessment and short-term power forecasting.





# 1    Introduction

The "European Green Deal" targets on making Europe climate neutral in 2050. To reach this goal, a decarbonisation of the
energy system is necessary, where increasing offshore wind production will play a key role European Commission (2019).
With planned installed capacities of 30 GW in 2030, 40 GW in 2035 and 70 GW in 2045 the new German government has
three long-term goals for offshore wind farm expansion in their coalition agreement (SPD, Bündnis 90 / Die Grünen, FDP,
2021). As areas are limited, a most effective arrangement of wind farms is desired. Thus, offshore wind farms are typically
grouped in wind farm clusters, which can be composed of several hundreds of turbines.

A recent study of large-scale wind farm expansion in the German Bight suggests that the yield decreases with an increasing
density of installed capacity (Agora Energiewende, 2020). The comparatively smooth surface and less turbulent conditions
benefit the formation of far-reaching wakes. Thus, cluster wakes and their impact on the power production have received in-
creasing attention over the past years, applying different measurement and modelling techniques (e.g Christiansen and Hasager,
2005; Li and Lehner, 2013; Hasager et al., 2015b; Djath et al., 2018; Platis et al., 2018; Siedersleben et al., 2018; Ahsbahs
et al., 2020; Cañadillas et al., 2020; Nygaard et al., 2020; Schneemann et al., 2020; Cañadillas et al., 2022).

Far-reaching wind farm and cluster wakes were observed in satellite synthetic aperture radar (Djath and Schulz-Stellenfleth,
2019) also in combination with scanning lidar (Jacobsen et al., 2015; Schneemann et al., 2020) and dual-Doppler radar (e.g.
Ahsbahs et al., 2020) as well as airborne measurement data (Platis et al., 2018). Offshore wind farm wake deficits last par-
ticularly long under stable conditions, but far-reaching wakes were also found under neutral and weakly unstable conditions
(e.g. Djath et al., 2018; Platis et al., 2020; Schneemann et al., 2020). For very constant wind directions aligning with stable
atmospheric stratification, Cañadillas et al. (2020) report wake lengths exceeding 80 km. The magnitude of the wake deficit far
downstream ranges from 25 % to 41 % of the wind velocity for a distance of 24 km and 21 % at 55 km downstream (Schnee-
mann et al., 2020). In the German Bight, conditions that benefit far-reaching wakes are expected to have a probability of about
5 % (Platis et al., 2018). The occurrence of stable conditions benefiting far-reaching wakes is depending on the wind direction.
Therefore, the occurrence of stable conditions in the main wind direction sector should be taken into account for the layout
optimization of wind farms (Emeis, 2018; Platis et al., 2018; Cañadillas et al., 2020).

It is well established that the wake of wind farms and wind farm clusters can impair the power production of downstream
wind plants. Nygaard and Hansen (2016) report a decreasing power production in the front rows of a downstream wind farm.
Meanwhile, Schneemann et al. (2020) observed an area of reduced power production in the center of the wake, while speed
up effects occurred at the edges during partial cluster wakes conditions. While the length of wakes has been studied on the
scale of wind farms and wind farm clusters, their propagation in a homogeneous atmospheric flow has been studied only on
the single turbine to farm scale. Few studies have focused on the wake deflection imparted by Coriolis force effects: van der
Laan et al. (2015) investigated the phenomenon using computational fluid dynamics (CFD) and concluded that considering
the Coriolis force affects the prediction of farm losses due to wake shadowing. In a later work, van der Laan and Sørensen
(2017) explained the clockwise turning of wind farm wakes by the entrainment of wind veer from above the farm. This was
studied in more detail by Gadde and Stevens (2019) using large-eddy simulations (LES) with the result that at the inflow of the





wind farm a counterclockwise rotation in the wind flow is observed as a consequence of wind farm blockage and wind veer. Further downstream, the downward mixing of veered flow from above the wind farm causes a clockwise deviation that directly impacts the trajectory of the farm wake (Eriksson et al., 2019; Gadde and Stevens, 2019). Furthermore, Eriksson et al. (2019)
investigated the impact on the power production of a downstream wind farm, showing a difference of 3 % in power production for a single flow case when including the terms associated with the Coriolis force.

The previously summarized studies show that it is relevant to further examine the advection of wakes on scales of several kilometers - the mesoscale - and account for it in engineering wake models as these are commonly used for yield estimations prior and after wind farm construction. Despite the higher uncertainty when compared to CFD simulations, they can assess a
large variety of different inflow conditions in a very computationally efficient manner (Machefaux et al., 2015).

A grid-based approach to consider wake advection in heterogeneous background wind fields in the engineering model context was developed for FLORIS (FLOw Redirection and Induction in Steady State) (NREL, 2020). In the model the flow is calculated on a mesh, rotated around the center of the flow field according to the local wind direction prior to the wake calculation. A backward rotation projects the wind direction changes of the background flow onto the wake (Farrell et al.,
70  2020).

In this study, we propose a grid-less approach which enables to use all the computational benefits of grid-less wake modelling. The model is coupled with background wind fields from a mesoscale numeric weather model for taking into account wind direction changes on large scales. The difference on calculated annual energy production (AEP) between the model and a simplified model solution for heterogeneous background flow is compared for a real example case of two large wind farm
clusters in the German Bight.

The recently published approach by Lanzilao and Meyers (2022) applies a methodology for deflecting the turbine wakes along the streamlines from a heterogeneous background flow field. In this study, we suggest the coupling of engineering wake models with mesoscale background wind fields for taking into account mesoscale wind direction changes and thus apply an approach similar to Lanzilao and Meyers (2022) but on larger scales. This allows us to demonstrate how the method works and
to discuss its implications when simulating for AEP calculation or site assessment.

The main objectives of this study are (1) to investigate the relevance of mesoscale wind direction changes taking the German Bight as example, (2) to propose a modelling approach at the scale of cluster wakes, (3) to implement this approach into an engineering model framework, and (4) investigate this approach on the scale of cluster wakes.

Section 2 describes the methods used for investigating the importance of mesoscale wind direction changes. Furthermore, the
developed wake propagation model and the simulations performed for its validation are described. Section 3.1 then introduces the engineering wake model suite *flappy* developed at the Fraunhofer IWES with which the simulations were conducted. Section 4 briefly describes the data used for evaluating the importance of mesoscale wind direction changes, as well as the simulation of the streamline model and its comparison with SAR data. In Sect. 5, the results are presented and discussed, split into the investigation of the wind direction changes and the evaluation of the cluster wake propagation. Finally, the findings are
summarized in Sect. 6.



## 2 Methods

In this section, we present the methods to investigate the relevance of mesoscale wind direction changes for different spatial (Sect. 2.1.1) and temporal scales (Sect. 2.1.2). A mechanism to simulate a wake propagation following the mesoscale wind direction changes in an engineering model context is introduced (Sect. 2.2).

### 2.1 Investigation of mesoscale wind direction changes

One aim of this study is to examine under which conditions large wind direction changes occur on the mesoscale. This evaluation is based on mesoscale model data originating from the New European Wind Atlas (NEWA) which are described in Sect. 4.1.

#### 2.1.1 Spatial scales

For the investigation of the impact of mesoscale wind direction changes, we first define three different scales determined by the distance of prominent offshore locations (wind farms, an offshore substation and two meteorological masts) in the German Bight: the *intra-cluster* scale which corresponds to the distance between the offshore substation DolWin1 and the met mast FINO1, the *inter-cluster* scale defined by the distance between the wind farms Deutsche Bucht (DBU) and Global Tech I (GT I) and the *German Bight* scale by met masts FINO1 and FINO3. Figure 1 shows the locations and distances between these locations.

The comparison of wind directions at different spatial scales was performed using mesoscale model timeseries of wind direction and speed at the grid points named above over a period of 30 years. The difference in wind direction between the two time series was calculated for each time step. To examine the dependency on the wind speed, they were then binned by the wind speeds at a reference location ($\mathrm{WS_{REF}}$):

$$\Delta\mathrm{WD}(bin_\mathrm{i}) = \Delta\mathrm{WD}(\mathrm{WS_{REF}} \in bin_\mathrm{i}) \tag{1}$$

with $i \in n$ and $n$ as the number of bins. A bin width of $2.5\,\mathrm{m\,s^{-1}}$ was chosen, as this was found to depict the dependency of $\Delta\mathrm{WD}$ on the wind speed while still reducing the number of bins to a reasonable amount. For each wind speed bin, the median, quartiles and the percentiles of wind direction differences were calculated.

In a second step, the whole German Bight area was investigated to examine the horizontal patterns of wind direction changes for a climatology of 30 years. The wind direction data were again binned by wind speed at the reference location and the mean values of the yearly median, upper and lower quartiles of $\Delta\mathrm{WD}$ at each grid point were calculated. The wind direction change was calculated as a difference in wind direction between each grid point $P_\mathrm{j}$ and the reference point, $P_\mathrm{REF}$ and the statistical parameters described above were calculated:

$$\Delta\mathrm{WD}(P_\mathrm{j}, bin_\mathrm{i}) = \Delta\mathrm{WD}(P_\mathrm{j}, \mathrm{WS_{REF}} \in bin_\mathrm{i}) - \mathrm{WD}(P_\mathrm{REF}, \mathrm{WS_{REF}} \in bin_\mathrm{i}) \tag{2}$$

In order to account for the distances to the reference location, the wind direction changes were normed to $100\,\mathrm{km}$ using the great circle distance between the $P_\mathrm{j}$ and $P_\mathrm{REF}$.



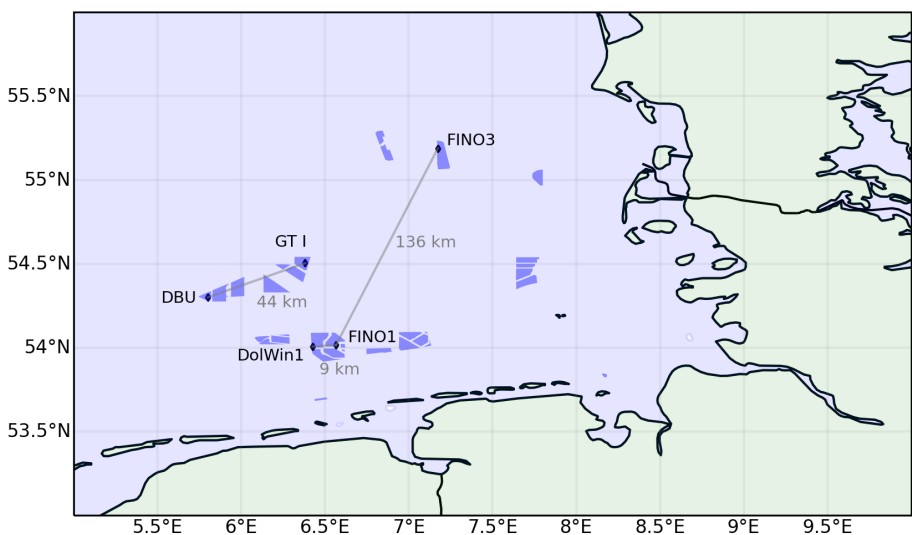

**Figure 1.** Different scales in the German Bight: *Intra-cluster* scale: DolWin1 - FINO1, *Inter-cluster* scale: Deutsche Bucht (DBU) – Global Tech I (GT I) and *German Bight* scale: FINO1 – FINO3. For the evaluation, the NEWA grid points closest to the selected sites were taken. The wind farms that are currently operational and under construction are marked in blue (data from Bundesamt für Seeschifffahrt und Hydrographie (2020)). The coastline and border data originate from the Natural Earth dataset (Natural Earth, 2021).

The area in Fig. 2 marked in grey was chosen for analyzing mean wind direction changes over the whole German Bight, as it excludes onshore sites. This area has a minimum distance to the nearest coastline of 5.3 km southwards and 7.7 km eastwards. After the mean of the yearly percentiles was taken for each grid point, the mean of the grid points in the offshore area was calculated.

### 2.1.2 Temporal scale: pressure systems

Wind direction changes are expected to be more pronounced on shorter time scales during the passage of pressure systems. Cyclones are usually transient and characterized by a strong curvature of the isobars. The passage of their frontal system is related to a drop in air pressure and high wind speeds (Bott, 2016). Therefore, the wind direction changes were investigated for single cyclone events by choosing time frames of two days around time periods where wind speeds above $15\,\mathrm{ms}^{-1}$ were persisting at FINO1 for each event. Anticyclones in contrast are often prevailing over longer time periods and are often characterized by relatively constant high atmospheric pressure at the surface. Thus, the examined high pressure situations were filtered by sea level pressure constantly exceeding 1020 hPa at FINO1. These conditions were found to last about four days in the selected examples. It has to be noted that in this part of the study, particularly pronounced pressure systems were investigated to examine extreme situations.

As before, the analysis was based on mesoscale model data. The wind direction at the mesoscale grid point closest to FINO1 was subtracted from the one at each grid point in the German Bight, and the resulting direction changes visualized in maps.



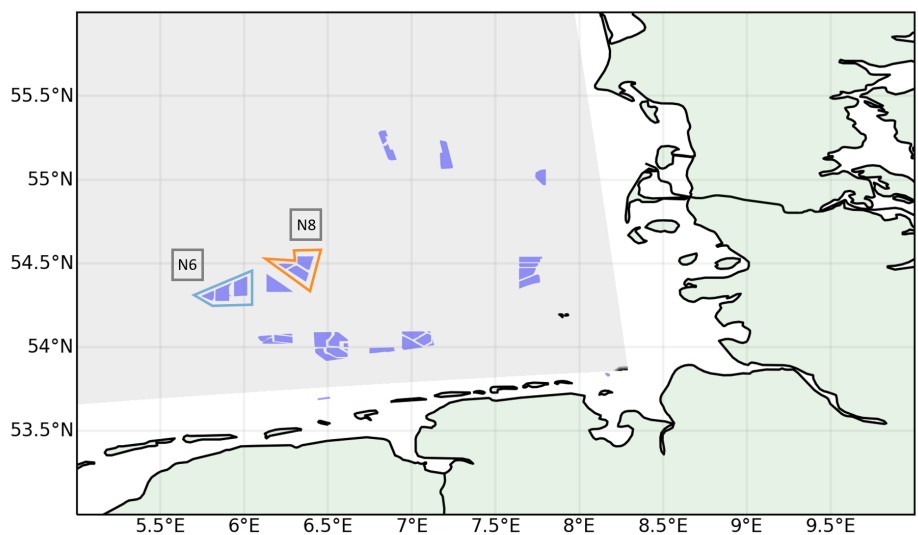

**Figure 2.** German Bight, selected area for calculating mean wind direction changes. The wind farms currently commissioned and consent authorized in the German Bight are marked in blue (data from Bundesamt für Seeschifffahrt und Hydrographie (2020)). The wind farms of the N6 cluster are framed in blue whereas the ones of the N8 cluster are framed in orange. The coastline and border data originate from the Natural Earth dataset (Natural Earth, 2021).

Furthermore, the mean wind direction change $\overline{\Delta\mathrm{WD}}$ over these above-mentioned time frames was calculated for different spatial scales as shown in Fig. 1.

## 2.2 Implementation of mesoscale wind direction changes into engineering models

In order to capture the mesoscale wake advection in the background flow, we determine the coordinate transformation from the global Cartesian frame of reference to the wake frame of reference for each wake-causing turbine. The wake frame is defined as a right-handed orthonormal coordinate system whose first axis follows the streamline curve that is originating at the rotor center.

For the wake calculations, our method considers any time step in the mesoscale time series as an independent heterogeneous steady-state flow field. Denoting the latter as $\boldsymbol{U}(\boldsymbol{x})$, the path $\boldsymbol{r}(t)$ of a probe particle moving in this background flow field is determined by the first order differential equation $d\boldsymbol{r}/dt = \boldsymbol{U}(\boldsymbol{r})$. For the formulation of the streamline, we choose a parametrization by the distance $s$ along the path rather than the time $t$, implying $ds = |\boldsymbol{U}|dt$. This yields the equation for the streamline curve $\boldsymbol{r}(s)$,

$$\frac{d\boldsymbol{r}}{ds} = \frac{\boldsymbol{U}(\boldsymbol{r})}{|\boldsymbol{U}(\boldsymbol{r})|} \ . \tag{3}$$



At any point of the streamline the normalized local tangent vector induces a right-handed orthonormal coordinate system with axes $\boldsymbol{e}_1$, $\boldsymbol{e}_2$, $\boldsymbol{e}_3$,

$$\boldsymbol{e}_1(s) = \frac{\boldsymbol{U}(\boldsymbol{r}(s))}{|\boldsymbol{U}(\boldsymbol{r}(s))|} , \qquad \boldsymbol{e}_2(s) = \boldsymbol{e}_z \times \boldsymbol{e}_1(s) , \qquad \boldsymbol{e}_3 = \boldsymbol{e}_z , \qquad (4)$$

where $\boldsymbol{e}_z$ is the Cartesian direction vector in $z$ direction. For an arbitrary point $\boldsymbol{x}$, we now define coordinates $(c_1, c_2, c_3)$ by

$$c_1(s) = s + (\boldsymbol{x} - \boldsymbol{r}(s)) \cdot \boldsymbol{e}_1(s) , \qquad c_2(s) = (\boldsymbol{x} - \boldsymbol{r}(s)) \cdot \boldsymbol{e}_2(s) , \qquad c_3 = \boldsymbol{x} \cdot \boldsymbol{e}_z . \qquad (5)$$

Equation (3) is then solved in discretized form for each rotor, with $\boldsymbol{r}(0)$ representing the rotor centre. For each location $\boldsymbol{x}$ Eqns. (4) and (5) are then computed at the nearest point of the resulting piecewise linear streamline. Finally, the wake model equations are evaluated based on the resulting coordinates.

## 3 Model and set-up

In this section, we introduce the Fraunhofer engineering model suite (Sect. 3.1) as well as its set-ups used for the simulations presented in this study (Sect. 3.2).

### 3.1 The Fraunhofer engineering model suite

All wind farm and wake calculation results in this work were obtained using[1] the *Farm Layout Program in Python* (*flappy*) developed by Fraunhofer IWES. This code is based on grid-less wake model superposition, and it has been optimized for numerous input states, like long-term time series or statistical distributions. Parallel multiprocessor runs are realized via the Ray library (The Ray Team, 2022; Moritz et al., 2018), and wind farm optimization problems can be solved based on an interface to pygmo (The pagmo development team, 2022). *flappy* has been recently applied to very different research questions, (Centurelli et al., 2021; Schmidt et al., 2021, 2022).

The philosophy behind *flappy* is that its core structure is widely vectorized and parallelized, but nonetheless the code is fully modular and easily extendable. Each wind turbine can be equipped with its own set of selected models. Minimally, this set contains a rotor model, a wake model, a wake frame model, and a turbine model that evaluates the thrust curve. Additionally, the wake superposition rules have to be specified.

During the calculation, the rotor model determines all rotor effective quantities, for example the rotor effective wind speed (REWS), based on one or more evaluation points. It is also responsible for handling partial wake effects. The wake model represents the wake behind a single isolated wind turbine. Various wind deficit and turbulence intensity wake models have been implemented into *flappy*, including numeric models that interpolate between tabulated fields from steady-state CFD simulations (similar to Schmidt and Stoevesandt, 2014, 2015).

The wake frame model provides the coordinate transformation from the global frame of reference to the wake frame of reference (Schmidt and Vollmer, 2020). The inflow wind vector as seen by the wake model in the wake frame is uniform and

---

[1] flappy version v0.5.2



parallel to the first axis. However, this does not necessarily have to be the case from the point of view of the global frame of reference, depending on the selected wake frame model. In fact, we make explicit use of curved coordinate systems following streamlines of the flow throughout this work, cf. Sect. 2.2. This choice models the transport of all wake effects along the background flow.

Turbine models can be added to the calculation as desired for variable calculation and manipulation. For example, a turbine model can evaluate the thrust and power curves of the turbine type in question based on the current REWS. Other models may be added for adding tabulated loads, or in order to realize curtailment, or for any other desired operation on the data during the calculation. The underlying modular structure makes *flappy* very flexible and applicable to a wide range of wind farm related calculations.

### 3.2 Study-specific set-up

To test the new streamline wake propagation model, we conducted both qualitative and quantitative comparisons against the more traditional baseline method. The latter method assumes a straight-line propagation of the wakes along the wind direction at the wake causing rotor.

For the qualitative comparison, situations with strong wind direction changes in the background flow have been selected to better highlight the differences between the models. To demonstrate the wake turning, the passage of a cold front with strong wind direction changes is simulated for the neighboring clusters N6 and N8. Additionally, single situations spotted through SAR imagery to be emblematic of the wake turning phenomenon were compared to *flappy* simulations spanning the whole German Bight.

In the quantitative analysis, the focus narrowed again on the N6 and N8 clusters to discuss how the new streamline method can impact either AEP computation, site assessment, or other typical applications of engineering models. The selection of these two farm clusters (compare Fig. 2) allows us to focus on the wind direction changes at an *inter-cluster* scale and how they affect cluster wake propagation. This scale is sufficiently big to observe such effects, but sufficiently small to avoid the vanishing of the cluster wake in the engineering models.

For calculating the velocity deficit in the wake, we rely on the Gaussian wake model of Bastankhah and Porté-Agel (2016). In this model, the dimensionless wake deficit outside the near wake length $x_0$ at any point $\boldsymbol{x} = (x, r)$ in the wake frame of reference is given as,

$$\delta(\boldsymbol{x}) = \frac{\Delta U(\boldsymbol{x})}{U_0(\boldsymbol{x})} = \left(1 - \sqrt{1 - \frac{C_T}{8\left(\frac{1}{\sqrt{8}} + k^* \frac{x-x_0}{d_0}\right)^2}}\right) \exp\left(-\frac{r^2}{2d_0^2\left(\frac{1}{\sqrt{8}} + k^* \frac{x-x_0}{d_0}\right)^2}\right) \tag{6}$$

Here, $x$ is the distance from the rotor center in stream wise direction and $r$ the radial distance in the wake frame coordinate. $\Delta U(\boldsymbol{x})$ is the difference between the wind velocity in the wake and the free stream velocity $U_0(\boldsymbol{x})$, $d_0$ the rotor diameter and $C_T$ the thrust coefficient. The wake expansion coefficient is expressed by a linear relation $k^* = k_{\text{TI}} \text{TI}^* + k_b$, where $\text{TI}^*$ is either the ambient turbulence intensity $\text{TI}_{amb}$ in the far wake or the local turbulence intensity $\text{TI}_{loc}$ in the near wake. The locus of the





transition between the near and the far wake region is given as

$$\frac{x_0}{d_0} = \frac{1 + \sqrt{1 - C_T}}{\sqrt{2}(\alpha \text{TI}_{loc} + \beta(1 - \sqrt{1 - C_T}))} \tag{7}$$

The model of Crespo et al. (1999) is applied to compute wake-added turbulence intensity $\text{TI}_{add}$. In any location, $\text{TI}_{loc}$ is given by the quadratic superposition of $\text{TI}_{amb}$ and the $\max(\text{TI}_{add}^i)$ with $i$ being all the wakes reaching the location of interest. The parameters $k_{\text{TI}}$, $k_b$, $\alpha$, and $\beta$ modulate the wake recovery, they require fine calibration in order to obtain reasonable results out of the model.

In our simulations, two different calibrations were considered. For the results of Sect. 5.2.1, the wake model parameters were adjusted to minimize wake recovery to achieve slowly decaying wakes. The near wake length was set to zero, $k_{\text{TI}} = 0.05$ and $k_b = 0.0$. This allowed for a better visualization of wake propagation in the qualitative comparison.

Such a set of parameters exacerbates unrealistically the power loss due to wake effects within the farm, and it is not advisable for computing a meaningful energy yield of the two clusters. Therefore, for the quantitative analysis, the parameters were calibrated using an optimization procedure involving the supervisory control and data acquisition (SCADA) data of one of the wind farms in the wind farm clusters considered. The procedure, here not presented for brevity, determined the coefficients $k_{\text{TI}} = 0.23$, $k_b = 0.003$, $\alpha = 1.4$, and $\beta = 0.077$ to guarantee the best agreement between the engineering model and the real production data.

For the specific implementation of the new streamline method, presented in Sect 2.2, we applied a discrete method to determine the streamline curves within the flow snapshots. The step width is chosen as $\Delta s = 500\,\text{m}$ as this value represents a good compromise between accuracy and computational time, given that the horizontal discretization of the input (background) flow field domain is of 3 km (see Sect. 4.1).

The choice of a wake superposition method is the last topic to discuss concerning the set-up of the engineering model. As the streamline wake model is meant for dealing with heterogeneous inflow conditions, we decided to consider in addition to the linear superposition method, which is typically advised when accounting for wake-added turbulence intensity (Niayifar and Porté-Agel, 2015), also the novel wake superposition method presented by Lanzilao and Meyers (2020). The purpose of this new method is, in fact, to cope better with non-homogeneous wind fields.

In the linear superposition method, the wake deficit $\delta_i(x)$ is evaluated at a point $x$ for all turbines $i \in N_t$ with $N_t$ as the number of turbines. This is then aligned to and superposed onto the background wind vector $\boldsymbol{U}_0(\boldsymbol{x})$ in the point $\boldsymbol{x}$,

$$\boldsymbol{U}(\boldsymbol{x}) = \boldsymbol{U}_0(\boldsymbol{x}) - \left( \sum_{i=1}^{N_t} \delta_i(\boldsymbol{x}) \, U_{\text{REWS,i}} \right) \boldsymbol{n}(\boldsymbol{x}) . \tag{8}$$

Here, the wake deficit is modulated by the rotor effective wind speed at the wake causing rotor $U_{\text{REWS}}$. $\boldsymbol{n}(\boldsymbol{x}) = \frac{\boldsymbol{U}_0(\boldsymbol{x})}{|\boldsymbol{U}_0(\boldsymbol{x})|}$ is the local directional unit vector and orients the wake deficit to the local wind direction. Additionally, to avoid a negative velocity to be computed in velocity fields with large wind speed gradients the wake deficit is limited to not overcome the local background wind speed at the point $\boldsymbol{x}$. The second superposition method considered is the one described in Lanzilao and Meyers (2020):



$$\boldsymbol{U}(\boldsymbol{x}) = U_0(\boldsymbol{x}) \left( \prod_{i=1}^{N_t} [1 - \delta_{\mathrm{i}}(\boldsymbol{x})] \right) \boldsymbol{n}(\boldsymbol{x}) \tag{9}$$

This method rescales the local undisturbed velocity magnitude, $U_0(\boldsymbol{x})$, with a coefficient derived as the product of the percentage reduced velocity caused by any wake reaching the point of interest. In this way, the resulting perturbed flow field conserves
the original wind direction distribution. In the rest of the manuscript this method will be referred to also as "wind product".

## 4  Data

For the evaluation of wind direction changes in the German Bight as well as meteorological input data for the simulations, the mesoscale atlas of the New European Wind Atlas (NEWA) has been used. Furthermore, for the qualitative evaluation of the simulation results, Synthetic Aperture Radar (SAR) images have been taken.

### 4.1  NEWA data

The mesoscale atlas of the NEWA spans a time period from 1989 to 2018 with a temporal resolution of 30 minutes. It has been generated using the Weather Research and Forecasting model (WRF). Wind speed and direction as well as air temperature are provided as three-dimensional parameters at 50, 75, 100, 150, 200, 250 and 500 m. Several other parameters such as the 2 m-temperature, pressure, inverse Obukhov length or the 10 m-wind are given for a single height. The horizontal resolution
is 3 km (Dörenkämper et al., 2020; Hahmann et al., 2020). The air density was calculated from the temperature at the height levels and the surface pressure corrected for height. Furthermore, the stability corrected TI was calculated at 100 m and derived from the inverse Obukhov length as described in Emeis (2010) and Peña and Rathmann (2014).

### 4.2  SAR data

Synthetic Aperture Radar data were used for a qualitative evaluation of the wake deflection mechanism. The SAR satellite emits
the radar signals which interact with the ocean surface. Back to the SAR system the received signals are mainly explained by Bragg scattering. SAR system captures the change in the sea surface roughness caused by the perturbation of wind at the surface through the capillary waves and modulates the normalized radar cross-section (NRCS, $\sigma_0$). The near-surface wind field can be derived using geophysical model function (GMF Portabella, 2002; Hersbach et al., 2007). The GMF relates the NRCS to the sea surface wind speed and the wind direction. This study uses C-band Sentinel1A/B SAR data covering the year 2018, which
are generated by the Copernicus program, the European Commission's Earth Observation Program (ESA, 2000-2020). The collected SAR images are from the Interferometric Wide (IW) swath mode with a spatial resolution of 20 m, which provides a fine structure of the wind farm wakes. Schneemann et al. (2015) suggest the use of radar satellite scans to study the flow fields at the scale of wind farm wakes. Hasager et al. (2015a) argue that SAR data are valuable for investigating the cluster wakes, as other form of measurements in the far-wake region are lacking.





## 5 Results and discussion

In this section, we first present and discuss the results concerning the importance of mesoscale wind direction changes in the German Bight (Sect. 5.1). Afterwards, we show simulation results from the new wake advection mechanism (Sect. 5.2). Additionally, we discuss the limitations of this method (Sect. 5.3).

### 5.1 Importance of mesoscale wind direction changes

Here, we first present results from mesoscale wind direction variations on different spatial scales, followed by findings on the impact of single high and low pressure systems.

#### 5.1.1 Spatial scales

For the different spatial scales, the distances between the locations presented in Fig. 1 are investigated. For the following analysis, we consider three different wind speed ranges: below cut-in wind speed ($0$ - $2.5\,\mathrm{ms^{-1}}$), the partial load range ($2.5$ - $14\,\mathrm{ms^{-1}}$) and above rated wind speed ($> 14\,\mathrm{ms^{-1}}$). Those ranges originate from the technical specifications of the different turbine types currently (winter 2022) operational in the German Bight.

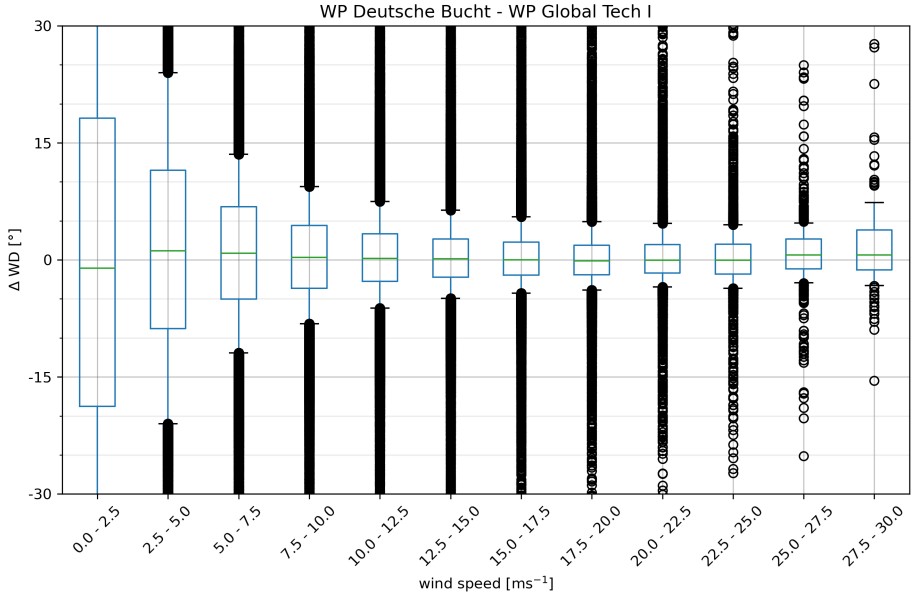

**Figure 3.** Wind direction changes between the locations of the wind farms Deutsche Bucht and Global Tech I (distance $\approx 44$ km *inter-cluster scale*) binned for the wind speed at the reference location (here: FINO1) obtained from 30 years of NEWA data (1989-2018). Green: median, blue box: quartiles, whiskers: 10 % and 90 % percentile, circles: outliers.



Especially for low wind speeds which are associated with high thrust coefficients, the variability in $\Delta$WD is large while reducing towards higher wind speeds. This is exemplarily shown for the *inter-cluster* scale as a boxplot in Fig. 3. It can be observed that the median of $\Delta$WD reduces towards the upper partial load range while increasing again towards cut-out wind speed. Overall, the median varies between about $\pm 1°$. However, in 50 % of the cases as denoted by the quartiles in Fig.3 considerable wind direction changes of more than 9° in the lower operational range (see e.g. 2.5-5.0 ms$-1$) can be found. Averaged wind direction changes for the different operational ranges are summarized in Tab. 1.

**Table 1.** Wind direction changes [deg] for the different operational ranges and spatial scales from the NEWA data.

| wind speed [ms$^{-1}$] | **intra-cluster scale** | | |
|---|---|---|---|
| | lower quartile | median | upper quartile |
| 0.0 - 2.5 | -7.2 | -0.2 | 7.0 |
| 2.5 - 14.0 | -1.5 | -0.1 | 1.3 |
| 14.0 - 30.0 | -0.9 | -0.2 | 0.5 |
| wind speed [ms$^{-1}$] | **inter-cluster scale** | | |
| | lower quartile | median | upper quartile |
| 0.0 - 2.5 | -18.8 | -1.1 | 18.2 |
| 2.5 - 14.0 | -3.8 | 0.4 | 5.0 |
| 14.0 - 30.0 | -1.9 | 0.0 | 2.2 |
| wind speed [ms$^{-1}$] | **German Bight scale** | | |
| | lower quartile | median | upper quartile |
| 0.0 - 2.5 | -48.3 | 8.3 | 59.0 |
| 2.5 - 14.0 | -9.0 | 1.1 | 12.8 |
| 14.0 - 30.0 | -6.3 | -1.4 | 3.7 |

It becomes clear that with increasing distance, the mesoscale direction changes become more relevant. However, also in the intra-cluster scale, 50 % of the cases exceed about 1.5 ° on average in the partial load range. Considering that far-reaching cluster wakes have been measured at distances between the *inter-cluster* and the *German Bight* scale, the focus needs to be laid at these scales concerning cluster wake deflection. Here, wind direction changes are commonly exceeding about 5° and about 10° in operational conditions, where direction changes reach twice the magnitude in the lower operational wind speed range of 2.5 - 5.0 ms$^{-1}$ (not shown). Even though the wake associated wind speed deficit does not impact the power production above partial load range, the mechanical loads due to the increased turbulence in the wake can reduce the turbines´ lifetime and thus are of relevance.

In the next step, we evaluated the mean large-scale horizontal patterns of wind direction changes for the 30 year NEWA period (1989-2018) as shown in Fig. 4. The mean was taken of the yearly median, upper and lower quartile binned by wind speed at the reference location. We calculated the direction change with respect to the reference location FINO1 and normed the values by 100 km. The wind direction changes for all wind speeds (see Fig. 4 - upper row) show a dipole pattern for the



**Figure 4.** Wind direction changes with reference to FINO1, normed for 100 km for different wind speed ranges. The left and right plots show the mean of the annual lower and upper quartiles respectively. The center plots shows the mean of the yearly medians for all wind speeds (upper row), the wind speed range 2.5 - 5.0 ms$^{-1}$ (middle row) and 12.5 - 15.0 ms$^{-1}$ (lower row). Note the different color scales for the wind speed ranges. The coastline and border data originate from the Natural Earth dataset (Natural Earth, 2021).



mean of the medians, with positive $\Delta$WD on the seaward side of the reference location and negative ones towards the coastal
regions. Negative direction changes can be interpreted as clockwise turning winds (veering), while positive values correspond
to anti-clockwise turning (backing). A coastal effect can be observed at the eastern and southern coastlines of the German
Bight.

Considering the area relevant for offshore wind farms (see Fig. 2), the median of $\Delta$WD ranges between $\pm 3\,°/100\,\text{km}$.
Nevertheless, the upper and lower quartiles show values reaching to $-23\,°/100\,\text{km}$ and $+24\,°/100\,\text{km}$ in this offshore region.
For the wind speed bin representing the lower partial load range (Fig. 4 - central row), the medians show a similar range while
the quartiles show much larger variation of the data, reaching up to $-46\,°/100\,\text{km}$ and $+68\,°/100\,\text{km}$. Towards the upper end
of the partial load range (Fig. 4 - lower row), this variation reduces to a range of $-17\,°/100\,\text{km}$ and $+13\,°/100\,\text{km}$. For wind
speeds above this, the range of wind direction changes increases slightly again. With increasing wind speeds, the dipole pattern
of $\Delta$WD becomes more distinct (Fig. 4 - lower row) which aligns well as the main wind direction at FINO1 shifts to west-
southwest for higher wind speeds (see e.g. Jimenez et al., 2007). This is related to common routes of cyclones over the North
Sea (van Bebber, 1891; Hofstätter et al., 2016).

The presence of coastal effects could be observed mostly for moderate to high wind speeds (see e.g. Fig. 4). This results in
comparably large direction changes, which reach up to some tens of kilometers offshore. One explanation is the sudden change
in surface roughness which leads to a new balance of Coriolis, pressure gradient and frictional forces and thus a turning of
the wind direction at the coastline. In case of sea surface temperatures colder than the advected air mass, an induced stable
stratification develops, which can in the end lead to the formation of low level jets (Emeis, 2018).

**Table 2.** Mean values of the percentiles for the wind direction changes in $°/100\,\text{km}$ with reference to FINO1. The values are calculated as
an average of all grid points in the offshore regions of the German Bight (compare Fig. 2).

| wind speed [ms$^{-1}$] | mean of lower quartile | mean of median | mean of upper quartile |
|---|---|---|---|
| all | -7.1 | 0.5 | 7.7 |
| 0.0 - 2.5 | -38.9 | -2.8 | 35.1 |
| 2.5 - 5.0 | -20.8 | -1.8 | 17.4 |
| 5.0 - 7.5 | -11.7 | -0.5 | 10.4 |
| 7.5 - 10.0 | -7.1 | 0.3 | 7.5 |
| 10.0 - 12.5 | -5.0 | 0.5 | 6.0 |
| 12.5 - 15.0 | -3.4 | 0.9 | 5.3 |
| 15.0 - 17.5 | -2.6 | 1.1 | 5.2 |
| 17.5 - 20.0 | -1.9 | 1.4 | 5.3 |
| 20.0 - 22.5 | -1.4 | 1.8 | 5.8 |
| 22.5 - 25.0 | -0.8 | 2.7 | 7.0 |
| 25.0 - 35.0 | 0.3 | 4.0 | 9.0 |



Table 2 shows the mean values of the binned wind direction changes per 100 km averaged for the selected area (gray area in
Fig. 2). It can be summarized that 50 % of all cases show on average a wind direction change of more than 7 ° per 100 km (first
row of Tab. 2). Focusing on the partial load range (2.5 - 14 ms$^{-1}$), the means of the lower and upper quartiles decrease with
increasing wind speed until the upper end of the partial load range. In the lower partial load range, 50 % of the data show wind
direction changes exceeding - 20.8 ° / 100 km and + 17.4 ° / 100 km.

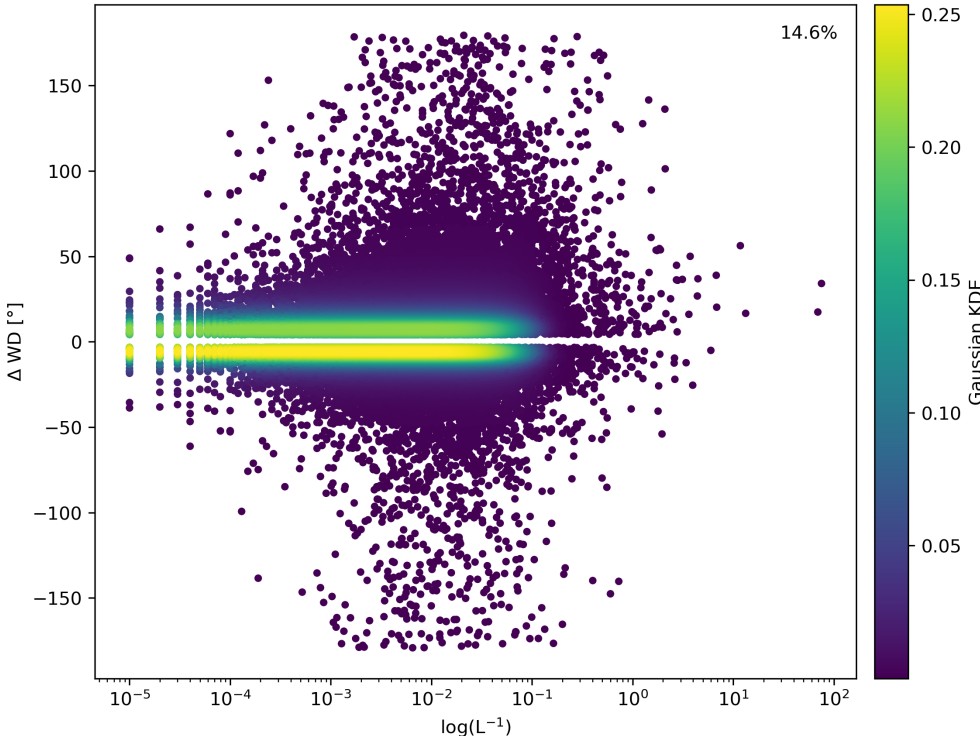

**Figure 5.** Wind direction difference (ΔWD) at the *inter-cluster* scale (DBU - GT I) for neutral and stable atmospheric stratification. Only
values for wind speeds above 2.5 ms$^{-1}$ and ΔWD exceeding the lower and upper quartiles are shown, which results in 14.6 % of the data.
The color shows a Gaussian Kernel density estimation.

As aforementioned, far-reaching cluster wakes were found under both neutral and stable atmospheric stratification. These
conditions occur with a probability of 29.2 % at the location of the wind farm Deutsche Bucht, indicated by a positive inverse
Obukhov length. Only conditions with wind speeds exceeding 2.5 ms$^{-1}$ were considered to exclude wind speeds below cut-in
wind speed and thus irrelevant for wake formation. Figure 5 shows the magnitude and probability of wind direction changes
exceeding the quartiles of -3.2 ° and +4.2 ° at the *inter-cluster* scale (DBU - GT I). Conditions under which wind direction
changes exceeding the lower and upper quartiles align with atmospheric stratification that benefits far-reaching wakes have a
probability of 14.6 %.





### 5.1.2 Temporal scale: low and high pressure systems

To investigate the situation during the passage of pressure systems over the German Bight, shorter time periods were selected for further investigation.

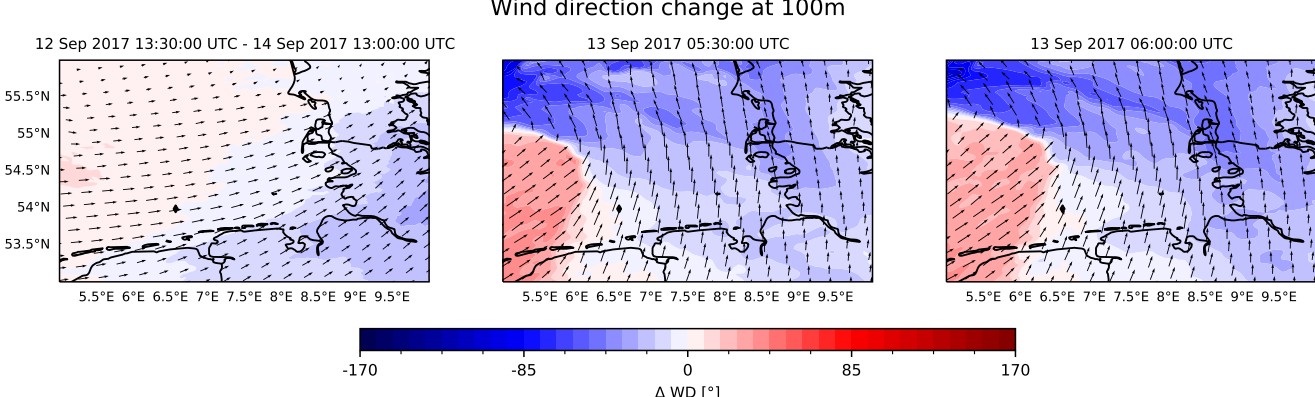

**Figure 6.** Wind direction changes during the passage of cyclone "Sebastian" in September 2017. The left subplot shows the median (time span of 48 h) of the wind direction changes with reference to FINO1, which is marked as a black diamond. The center and right plot show snapshots of the wind field while the cold front of "Sebastian" passed over FINO1. The black arrows indicate the wind direction and speed. The coastline and border data originate from the Natural Earth dataset (Natural Earth, 2021).

Figure 6 is an exemplary showcase of the wind direction changes happening as the frontal system of the cyclone "Sebastian" (in September 2017) passed over the German Bight. Typically, low pressure systems move quite fast from west to east in the northern mid-latitudes and their frontal systems are associated with strong and sudden wind direction changes. The left plot shows the median of the wind direction changes between each NEWA grid point in the German Bight and the one closest to the reference location FINO1 for a time frame of 48 h around the wind speed maximum.

During the early morning, the cold front passed over the German Bight, causing large wind direction changes (see center and right panel). The comparably large wind direction changes of the instantaneous velocity fields in these panels demonstrate the extent of wind direction changes on shorter time scales. The difference between the mean directions at DBU and GT I for the whole time period 12 September 2017 13:30:00 UTC to 14 September 2017 13:00:00 UTC is only about 2.3 ° while even smaller values occur for other cyclone passages. Contrasting the small mean difference in wind direction, peaks in $\Delta$WD on the *inter-cluster* scale were found to reach up to 50 ° and -80 ° during passage of the front shown in Fig. 6. This specific situation is further investigated with respect to large-scale wind direction changes in the following sections.

For anticyclones (not shown), comparably large wind direction changes were found for longer time periods. Over the course of four days, the difference in mean wind direction at DBU and GT I was e.g. 6.1 ° (06 May 2008 00:00:00 UTC to 09 May 2008 23:30:00 UTC) and 4.3 ° (25 June 2018 00:00:00 UTC to 28 June 2018 23:30:00 UTC). This highlights that anticyclones can cause wind direction changes at larger scales, persisting over longer time frames.





Specifically, the strong wind direction changes during the passage of cyclones are often less relevant for the power production
as the wind speed is mostly above rated wind speed. Nevertheless, the wake can reduce the wind speed at downstream wind
farms to $U < U_{\text{cut-out}}$. Furthermore, due to the increased turbulence in the wake, these conditions are relevant for load calculation
and therefore for the prediction of the turbines´ lifetime. Moreover, the warm air advection associated with the frontal system
of cyclones influences the atmospheric stability, as it often leads to stable stratification (Bott, 2016; Emeis, 2018) and thus to
increased wake lengths.

**5.2   Evaluation of the streamline wake model**

In this section, the capabilities of the new streamline wake model implemented in *flappy* are displayed in a series of com-
parisons. The wake propagation method where every single turbine wake evolves rectilinear downwind, oriented as the wind
direction at the wake causing rotor, is used as the baseline model. For simplicity, in the next paragraphs the new streamline
wake model and the baseline model are relabeled with the acronyms SWM and BLM, respectively.

**5.2.1   Cluster wake turning associated with pressure systems**

The wake effects of the clusters N6 and N8 (compare marked clusters in Fig. 2) have been simulated during the passage of the
cold front of the cyclone Sebastian (13 September 2017, compare Fig. 6).

Figure 7 shows the wind field at about hub height perturbed by the interaction with the wind farms. The three columns display
three relevant stages, i.e. instanced in time, of the phenomena. The BLM (upper row) shows unexpected wake behavior as soon
as the cold front approaches the N6 cluster, with the wakes of the leftmost turbines crossing the ones of the other turbines.
When the cold front passes over N6, the wake of this cluster reaches and impacts N8, despite the different wind direction to
the right of the front would prevent this from happening. In contrast, the lower panels of Fig. 7 show the wakes simulated with
the SWM. Already at the first time step, the new model describes the wake propagation in a more sounding manner, resolving
the wake crossing problem. Also, the following snapshots reveal a more realistic adaptation of the wakes to the heterogeneous
background flow.

**5.2.2   Qualitative comparison to SAR images**

The validation of the new SWM is initiated by a qualitative comparison to SAR imagery, offered in Figure 8. The lower panel
shows the normalized radar cross-section (NRCS) from Sentinel1A, acquired at 05:48 UTC. In the eastward direction, the
shadow areas and streaks downstream the wind farms indicate low backscatter and correspond to the region of reduced wind
speed that characterized the wakes. Quite long wakes and an evident wake turning induced by westerly winds that get a more
northern component in the southern part of the German Bight are visible. The upper panels of Figure 8 show the wind speed
at hub height above the mean sea level in the German Bight simulated for the 21st March 2018 at 06:30:00 UTC. All wind
farms installed in the area at that date have been considered. However, precise information on the turbine operation states were
missing. In this situation, the westerly wind is not particularly strong, and thus the turbines operate at maximum thrust, offering

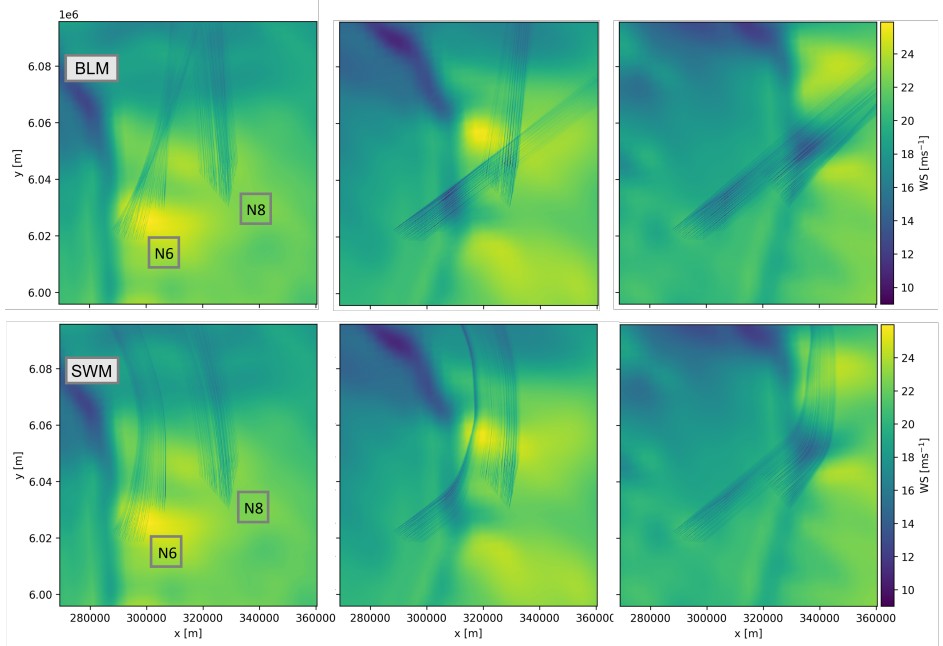

**Figure 7.** Cluster wakes of the N6 and N8 clusters in the German Bight, modeled with the baseline model (top) and the new streamline wake model (bottom). The situation chosen is the passage of a cold front captured in the mesoscale data of NEWA on 13 September 2017. The wind speed [ms$^{-1}$] at hub height is shown from left to right for the times: 05:00:00 UTC, 05:30:00 UTC, 06:00:00 UTC. The coordinates are given in UTM format (Zone 32U) with meters easting on the x- and meters northing on the y-axis.

long persisting wakes. The results of the simulations with the BLM (upper left panel) differ significantly from the one of the SWM (upper right panel). In comparison to the SAR image (bottom panel), the SWM model seems to gain a higher fidelity in the representation of the velocity field. The new model allows to correctly represent the distinct anticyclonic turning of the N3 cluster wake. Also, when looking at the Gemini cluster, we observe a much better agreement of the model with SAR when the SWM is used instead of the BLM.

Note that the displayed SAR was taken about 45 min earlier than the mesoscale model data. The different time in the models was chosen as it provided the best agreement with the SAR data, acknowledging that phase error up to several hours are a well-known phenomenon in mesoscale model data.

Contrasting the result presented above, in other situations analyzed the agreement between SAR and both modeling strategies was poor (not shown). However, this mismatch originates from a disagreement between SAR and the mesoscale input data.

Therefore, in the context of engineering modelling, these results do not diminish the attractiveness of the SWM against the BLM, as also the second would suffer from the same issue. Dörenkämper et al. (2020) found that on average, the mesoscale atlas of NEWA represents offshore wind conditions well.



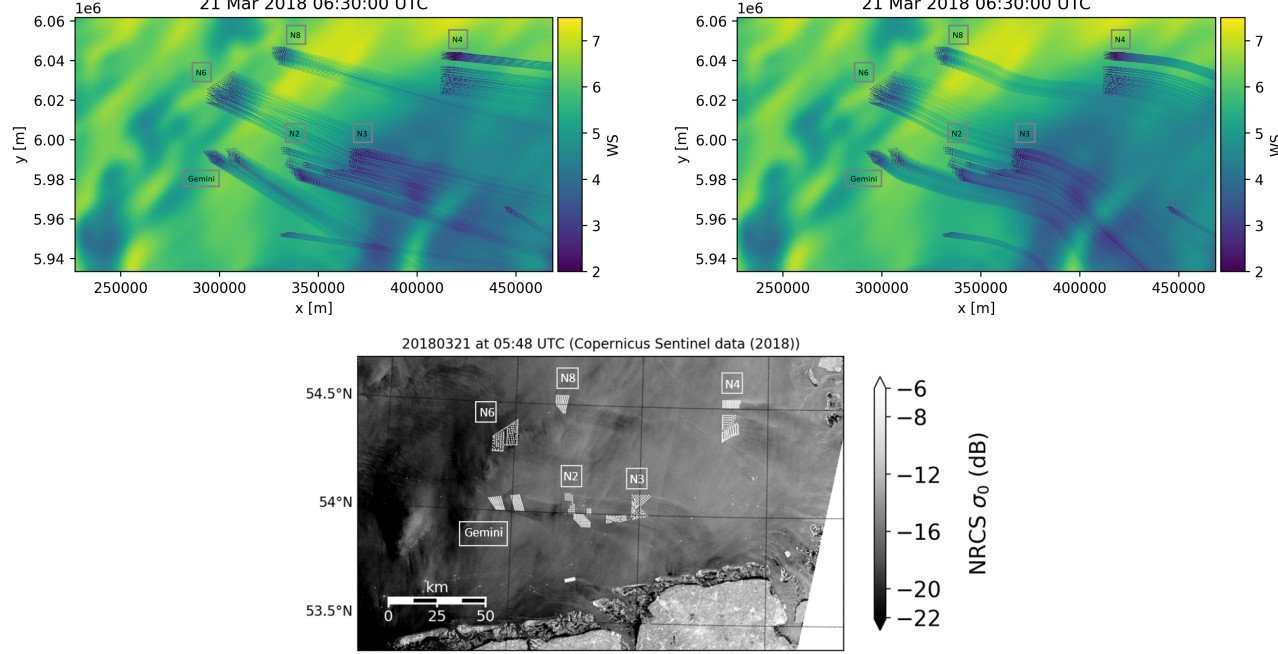

**Figure 8.** Upper: Wind speed [ms$^{-1}$] at hub height in the German Bight for 21 March 2018 at 06:30:00 UTC, simulated with flappy for all wind farms commissioned to that date. Left: baseline model with straight wake propagation, right: the new streamline wake model. Lower: SAR image from Copernicus Sentinel data (ESA 2000-2020) showing cluster wakes in the German Bight on 21 March 2018 at 05:48:00 UTC. The white dots show the wind turbines, the shading the backscatter value $\sigma_0$ of the normalized radar cross-section (NRCS).

Another point yet not discussed is the wakes' extent, significantly greater in the models than in the SAR image. As previously discussed, this was a design choice meant to avoid a full recovery of the wake to track their directional evolution throughout
the whole domain. However, a more careful choice of coefficients for the wake model may eventually grant better agreement not only in the directional wake propagation but also in their extent.

In conclusion, the comparison of Fig. 8 highlights the improvements in terms of the wakes´ directional propagation offered by the SWM. So far, SAR imagery has not been demonstrated yet as a reliable tool for quantitative wake model validation, as the data acquired have uncertainty of up to 2 ms$^{-1}$ (Hasager et al., 2020). However, the comparison with SAR highlights the
importance of considering mesoscale deflection of cluster wakes when the aim is to correctly assess how nearby wind farm clusters influence each other.

### 5.2.3   Impact on energy yield assessment

In this section, we discuss the differences across the SWM and BLM when applied to computing the energy yield at two wind farm clusters impacting each other. Assuming the proposed method as an improvement over the common BLM, these simula-



tions provide an estimate of the new model's potential to partly reduce and especially understand the sources of uncertainty in engineering wake model applications.

As before, the selected clusters for the study are the clusters N6 and N8 in the German Bight. The tool *flappy* is applied for calculating the power production at any time step in the last 10 years of the NEWA mesoscale data with the BLM and SWM model, respectively.

As a first analysis, the total energy yield of N6 and N8 when operating together is calculated. The simulation set-up adopted is described in detail in section 3.2. In addition to the two wake propagation methods, also two different superposition methods for the velocity deficit will be considered. Table 3 briefly summarizes the four simulations performed for this analysis.

**Table 3.** Simulations performed with the two wake propagation models and two wake superposition models in *flappy*.

| case | propagation model | superposition model |
|------|-------------------|---------------------|
| C1   | BLM               | wind linear         |
| C2   | BLM               | wind product        |
| C3   | SWM               | wind linear         |
| C4   | SWM               | wind product        |

The energy yield simulated over a period of 10 years shows that the SWM results are very similarly to the BLM results. The relative difference of the yield of both clusters is 0.01% and 0.004% for the superposition models wind product and wind linear,

respectively. However, the standard deviations of 1.21% and 1.05%, respectively, are suggesting that larger and non-negligible differences between the models occur at shorter time frames.

Most interestingly, the largest differences happen systematically according to wind direction and wind speed, as demonstrated in the heatmap of Fig. 9. According to the color coding, the relative difference between the models is greater for very low wind speeds or for wind speeds in the partial load range and wind directions where the clusters align. This can be explained

by effects on the *intra-cluster* as well as the *inter-cluster* scale. As already discussed in section 5.1, very low wind speeds are often associated with inhomogeneous wind fields. Therefore, significant wind direction changes are observed even at an *intra-cluster* scale. In this case, the models can provide different results even in situations where the two clusters should not interact through their wakes due to differing wake propagation within one cluster. For higher wind speeds, significant differences between the models are grouped around the wind directions at which one cluster is in the wake of the other. Once the wind speed

exceeds the rated speed, the differences progressively fade away as pitching control reduces the intensity of the wakes. This demonstrates the effect of the SWM on energy yield calculation on the *inter-cluster* scale.

The results presented so far suggest that the use of the SWM model could be well motivated for two main practices that engineering models enable: site assessment for new wind farms and short term forecasting for e.g. grid following control strategies. In both applications, it is important to guarantee that the engineering models compute with a sufficient level of

accuracy on short time scales.

The quantitative differences discussed so far are comprised of differences in the wake development, both on the scale of a single farm and the cluster wake scale. However, the underlying mesoscale data represent the flow field at the scale of cluster



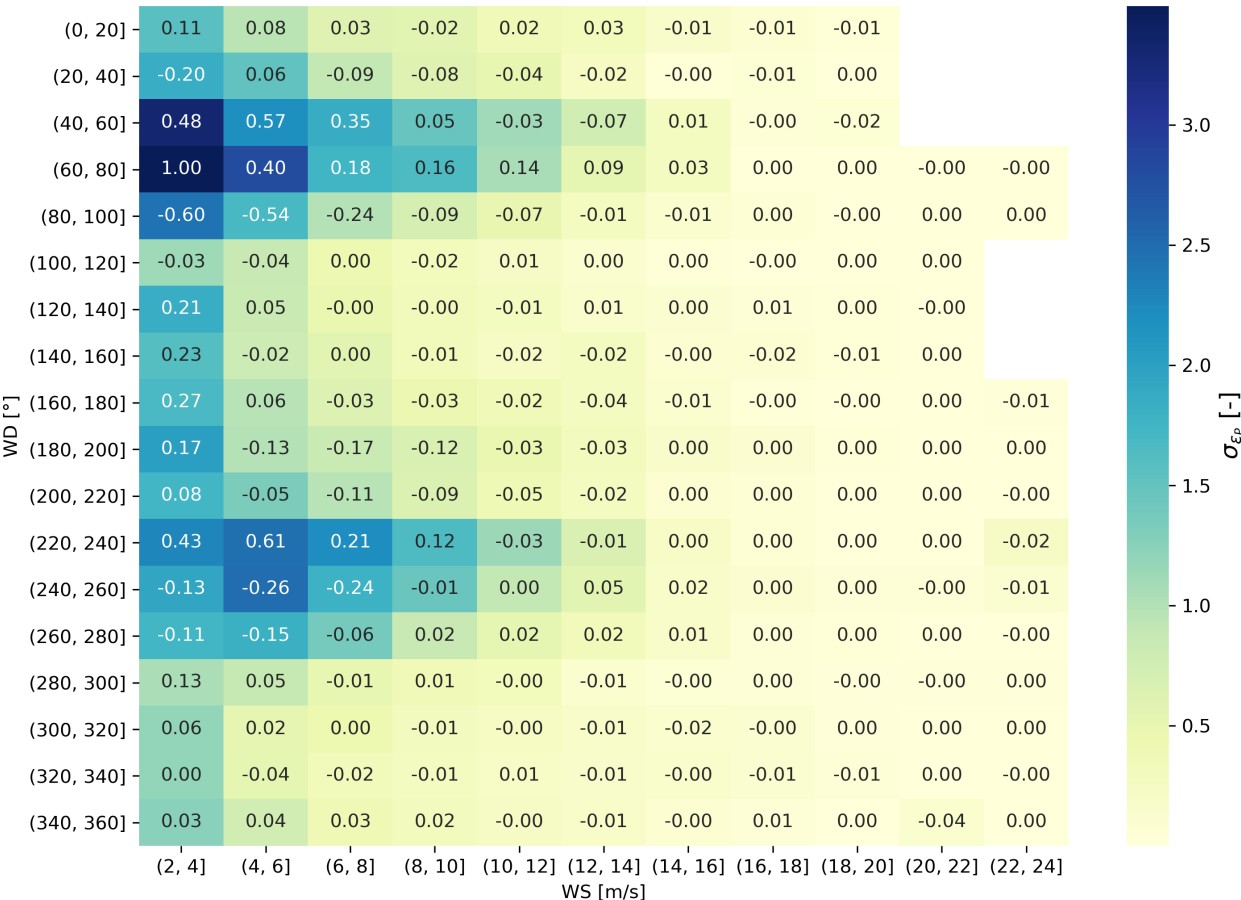

**Figure 9.** Heatmap of the standard deviation of the relative difference between the streamline wake model and the baseline model in the calculation of the energy yield of the N6 and N8 cluster. Here, the wind linear superposition was applied. Annotated values are the mean of the same value. The mean and the standard deviation are computed as , $\mu_{\epsilon_p} = \frac{1}{n}\sum_{i=0}^{n}(P_{i,\text{N6N8}}^{\text{SWM}}/P_{i,\text{N6N8}}^{\text{BLM}} - 1)$ and $\sigma_{\epsilon_p} = \frac{1}{n-1}\sqrt{\sum_{i=0}^{n}[(P_{i,\text{N6N8}}^{\text{SWM}}/P_{i,\text{N6N8}}^{\text{BLM}} - 1) - \mu]^2}$, respectively, with $n$ the last time step of the NEWA time series considered. The wind speed and direction considered for the bin is taken from easternmost turbine row of the cluster N6.

wakes well while lacking the accuracy to correctly describe the wind direction changes at a scale of a few rotor diameters. Thus, the uncertainty introduced by the effects within a cluster is isolated by performing the same set of simulations C1-4

(tab. 3) of N8 without N6. In these simulations, the energy yield of N8 is only affected by the *intra-cluster* effects. Therefore, the difference of energy yield of only N8 between these and the previous simulations gains the yield loss due to the cluster wake of N6. This value is again compared across the SWM and BLM models to estimate the expected uncertainty reduction offered by the SWM. The resulting difference for the two wake superposition models wind linear and wind product are 0.5% and 0.7%, respectively.





In conclusion, applying the proposed model reduces the uncertainty by up to 0.7% when estimating the interaction between neighboring wind farm clusters as N6 and N8. The applied wake model which was optimized on a single wind farm is likely underestimating the extent of cluster wakes. Considering this aspect in addition with ambitious plans for many more wind turbine installations, we claim that the benefit of using the new model will widen significantly.

### 5.3   Limitations of the streamline wake model

Several assumptions were made in the construction of the proposed streamline wake model, as in the context of engineering modelling any accuracy improvement should not compromise computational speed.

    The major assumption is that the wakes propagate according to the streamlines, rather than being a Lagrangian tracer. This means that any time step of the NEWA time series is considered as a stationary flow field that lasts for a sufficiently long time to allow the wake to propagate to their maximum extent. This assumption mainly stems from the coarse temporal resolution of

the mesoscale data of 30 minutes that would hinder the application of a true wake advection algorithm. However, due to this simplification all the various states remain independent of each other, allowing for a simplified algorithm and a more efficient parallelization.

    A further assumption taken is that the wakes cannot modify the background wind field, and thus the wake deficit is designed to always be aligned to the local background wind vector. Mostly in complex wind fields with convective behavior or strong

shear, this yields an unknown uncertainty. However, there are no sufficient studies of wakes in such situations to choose a more sophisticated approach.

    Further simplifications concern both the horizontal and the vertical description of the wake. As explained by van der Laan and Sørensen (2017) the direction in which a turbine wake propagates is influenced by both the wind veer and the Coriolis force. The wake modifies the vertical wind velocity profile, unbalancing the local equilibrium of forces. As a result, the wind

direction is locally changed in agreement with the Coriolis force. At the same time, the additional shear introduced by the wake deficit promotes mixing, especially from above. Due to this phenomenon, the local wind direction is also affected by the entrainment of the fluid layers above the wake that have a different wind direction due to the Ekman spiral. Gadde and Stevens (2019) found that the propagation of the wake of a wind farm in a homogeneous inflow is determined only by the entrainment of wind veer, and the effect of the Coriolis force is almost negligible. However, both of these phenomena are neglected in the

computation of the wake trajectory with the SWM model. Considering such local phenomena is expected to have a small gain in accuracy compared to the large uncertainty of mesoscale data in characterizing the flow at the turbine level. Additionally, the iterative calculation of the streamlines which would be necessary to include such effects would increase the calculation time.

### 6   Conclusions and outlook

Mesoscale wind direction changes are found to exceed 7° per 100 km in 50 % of all cases, referring to mesoscale data simulated

in the NEWA from 1989 to 2018, averaged for the German Bight. Particularly for wind speeds in the lower partial load range, large wind direction changes occur, with the quartiles reaching up to 20° per 100 km. Wind direction changes are found to



be specifically relevant on the horizontal scale of several tens to a hundred kilometers. Low and high pressure systems were found to lead to large wind direction changes which persist over shorter time periods for low and longer time periods for high pressure systems. Also, the evidence of coastal effects can be found in the wind direction changes to some tens of kilometers

offshore. Moreover, atmospheric conditions promoting far-reaching wind farm wakes were found to coexist with larger wind direction changes between wind farm clusters in 14.6 % of the cases. This favors the occurrence of far-reaching wakes with a strong turning on the scale of cluster to cluster interaction.

The magnitude and frequency of the wind direction changes observed in the mesoscale suggests that the uncertainty in the modelling of wind farms, especially for engineering models, can rise significantly with the number of farms simulated if the

wake propagation disregards changes in the flow direction. Thus, we present a new framework for engineering models that, given a 3D resolved wind field as input, accounts for wake turning with the background flow. This is realized by assuming the single turbine wake centerline to evolve as the streamline starting at the location of the wake causing turbine in the 2 D horizontal flow field located at hub height. Initial qualitative and quantitative results obtained through the method implementation into the Fraunhofer IWES wind farm modelling tool *flappy* are also discussed.

A qualitative comparison of the simulated flow fields around the wind farm clusters in the German Bight with SAR data shows that the new model can represent the flow with a far greater fidelity than the common baseline approach. However, the agreement of the new model with SAR is found to be inconsistent. We conclude that the accuracy of the model depends greatly on the accuracy of the underlying data (NEWA data in the discussed cases). However, it should consistently outperform the baseline approach that neglects wake turning.

In our quantitative studies, the new streamline wake model brought little to no benefit over the baseline when focusing on computing the energy yield of multiple wind farms across several years. However, considering more than two wind farm clusters and a wake model tuning favoring an accurate description of the cluster wakes, it is likely to obtain a different result. On the other hand, significant differences between the models are observed when computing the power output on shorter time frames or the impact of surrounding clusters on a site of interest. Given the positive feedback provided by the qualitative comparison to

SAR, we believe that the proposed method offers a reduction of uncertainty in engineering modelling over the baseline model. Therefore, as the coupling of engineering models and mesoscale data could change the paradigm of site assessment and offer the possibility of grid-following control strategies on a multi-wind-farms logic, we recommend the proposed model for these purposes.

Future development, partly already initiated, encompasses the attempt of validating the model with a combination of lidar

measurements and wind farm production data, and the comparison to other large-scale cluster wake modelling approaches such as mesoscale wake modeling.

*Code and data availability.*   The flappy model is available upon request. The mesoscale data used for computing the streamlines originates from the New European Wind Atlas which is publicly available via https://map.neweuropeanwindatlas.eu/.



*Author contributions.* AvB did most of the data curation and analysis and suggested a first version of a deflection mechanism. She drafted a
first version of the manuscript. AvB, GC and JS implemented the wake deflection model. JS is the key developer of the flappy model. LV was
involved in the derivation of the model setup and the calibration of the wake model. BD was mainly involved in the discussion of the SAR
validation. MD initiated the research, supervised the work and was mainly involved in the funding acquisition and the research discussion.
All authors contributed intensively to the writing and review of the manuscript.

*Competing interests.* The authors declare that there is no competing interest.

*Acknowledgements.* The results presented in this paper were derived in the framework of the X-Wakes (grant no. 03EE3008) project. The
X-Wakes project is funded by the German Federal Ministry for Economic Affairs and Climate Action (Bundesministerium für Wirtschaft
und Klimaschutz - BMWK) due to a decision of the German Bundestag. The simulations were partly performed at the HPC Cluster EDDY,
located at the University of Oldenburg (Germany) funded by BMWK (grant no. 0324005). We thank the NEWA consortium for making the
mesoscale data set available to the public. Maps have been generated using the cartopy (Met Office, 2010 - 2015) library for python.



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
