# Peer review of "An investigation of spatial wind direction variability and its consideration in engineering models"

_Wind Energy Science, 2022_

## Author Response (AR1)

**Reply to Reviewer 1 by Anna von Brandis et al**

**General comments:** This article has two main elements. The first element is a quite comprehensive examination of the spatial behaviour of wind direction change of German Bight region of the North Sea. The second is an assessment of a method to incorporate the spatial variation of wind direction in engineering wake models. Both elements are in my opinion very interesting to the readership, and worthy of investigation. The first element examining the behaviour I would say is the stronger part of the article. Several novel and, I imagine, "discussion starting" maps are figures are shown. Although it may be beyond the scope of the paper, it would be interesting to further examine some of the root causes for these interesting direction change properties and phenomena. The article also applies nicely the NEWA datasets, in this research work. The second element assessing the improvement gained by using the curvature of streamlines in engineering models, is the weaker part of the article. And I address several aspects of it in the specific comments.

Overall the criticism that I think needs to be addressed concerns, i) some more guidance about the implementation of the direction change in the models, ii) some more discussion of the detail about the differences in the estimates coming from BLM and SWM (Fig 9) (for example is the sign of the difference telling us anything about the behaviour? And if we expect around 14 % of the time long wakes that have curvature, is that consistent with the no significant difference in yield estimates), and iii) the term uncertainty is used a fair amount, but I don't think it is uncertainty that is discussed, no SCADA data is used in validating the models, and it is therefore difficult to estimate model errors. Instead, it is differences in estimates being discussed, and since the sign of the difference and the sign of the expected error is not really discussed, the reader is left unsure whether the differences are likely to lead to improvement or not.

Another issue is the assumption of stationarity of the streamlines for each application of the wake modelling. I have suggested some more discussion on this and thinking of advection timescales in the specific comments. For me, although this is an obvious disadvantage of the method, I think the method is certainly worthy of investigation, as it may provide good results within its range of validity. So I see this article as providing a good stepping stone in the research.

Overall, I'd say the article is suitable for publication if the general and specific comments are all addressed adequately in a revised manuscript.

⇒ *We thank the reviewer for the careful evaluation of our manuscript and the in general positive feedback and the constructive comments to the work presented. As the overall criticism mentioned above is also reflected in the specific comments, please find our detailed response separately for every point.*

**Specific comments:**

1. L9 "Pressure systems" is used quite frequently, I'd suggest "synoptic pressure systems" is a better term.

   ⇒ *Agreed and changed throughout the manuscript.*

2. L15 About "good agreement if the underlying mesoscale " . . . in the main part of the article it would be nice to support this a statement a bit more. For example show examples where the underlying mesoscale situation is not represented well.

   ⇒ *We disagree with this comment and think it is not meaningful to show situations that don't fit for several reasons: In several parts of the manuscript we demonstrate that there are situations where mesoscale turning is not of relevance, e.g. in the distributions of wind direction differences. In addition, in section 5.2.3 we show that there are situations that show only minor improvement when including the turning impact. For a validation of the NEWA atlas, different studies were published in recent year focussing on different aspects of the wind atlas, e.g.:*

   *Dörenkämper, M., Olsen, B. T., Witha, B., and Hahmann, A. N. et al.: The Making of the New European Wind Atlas – Part 2: Production and evaluation, Geoscientific Model Development, 13, 5079–5102, doi: 10.5194/gmd-13-507-2020, 2020.*

   *Hallgren, C., Arnqvist, J., Ivanell, S., Körnich, H., Vakkari, V., and Sahlée, E.: Looking for an offshore low-level jet champion among recent reanalyses: a tight race over the Baltic Sea, Energies, 13, 3670, doi: 10.3390/en13143670, 2020.*

   *Kalverla, P. C., Holtslag, A. A., Ronda, R. J., and Steeneveld, G.-J.: Quality of wind characteristics in recent wind atlases over the North Sea, Q. J. R. Meteorol. Soc., 146, 1498–1515, doi: 10.1002/qj.3748, 2020.*

*Luzia, G., Hahmann, A. N., and Koivisto, M. J.: Evaluating the mesoscale spatio-temporal variability in simulated wind speed time series over Northern Europe, Wind Energ. Sci. Discuss. [preprint], doi: 10.5194/wes-2022-33, 2022.*

*In the revised version of the manuscript we provide reference to these studies: "The SWM model agreed well to the SAR data in few other situations considered, however we observe cases where the agreement between SAR and both models was poor (not shown). For these cases, the mismatch is mostly related to a fundamental disagreement between SAR and the WRF mesoscale data used as input of the engineering models. While Dörenkämper et al. (2020) found that on average the mesoscale atlas of NEWA represents well offshore wind conditions, the generated wind fields may present local errors. Recently, the NEWA atlas is subject to validation activities for different applications (e.g. Hallgren et al., 2020; Kalverla et al., 2020; Luzia et al., 2022). At the same time, SAR imagery has not been demonstrated yet as a reliable tool for quantitative wake model validation, as the data acquired have uncertainty of up to 2 m/s (Hasager et al., 2020)."*

3. L19 When comparing the computed total energy yield, it was found no significant differences using the two methods. In the main text it would be good to learn more about the difference, the sign of the difference and discussion about how good the estimates are in the first place, i.e. since an engineering model is used over longer scales.

    ⇒ *Concerning Fig. 9 [now: Fig. 10], a new paragraph has been added to comment on the sign and value of the annotated numbers (mean of relative difference) that reads:*
    *"Looking at the mean relative difference between the models, the SWM predicts a larger energy production in all the occasion where the main wind direction is so that the cluster align, bins [40, 60] - [60, 80], [220 - 240]. Coherently with the fact that accounting for wake turning, the waked cluster will produce more, experiencing more "free-stream" velocity. At the same time, the SWM model computes a lower yield than the BLM when the wind direction has an offset. A trend of anti-clockwise turning in the wake is highlighted by the fact that the SWM computes the lowest yield with respect to the BLM for the bins [80, 100], [240, 260].".*
    *We don't see a reason to change the abstract (L19) as the overall conclusion is unchanged.*

4. L30 Important correction. The Agora study did not show that yield decreases with increased installed capacity. It did show the efficiency decreased, but yield still increases with additional capacity. Please correct.
    ⇒ *Of course, you are right, only the efficiency decreased. Thanks for pointing this out. Changed accordingly.*

5. L57 Do the authors make a distinction between wind veering and backing. I suppose veering is the usual direction of wind direction with increase height from surface in an Ekman spiral setting.
    ⇒ *We tried to be consistent throughout the document: veering is clockwise rotation of the wind. In the Ekman spiral setting the wind veers with height, yes. We changed the sentence to: "In a later work, van der Laan and Sorensen (2017) explained the clockwise turning of wind farm wakes by the entrainment of wind veer from above the farm due to the Ekman spiral."*

6. L64 About "Despite the higher uncertainty…" what is the context of this statement. For resource assessment, application of CFD may not necessarily give better results in practice because of the limitation of domain size, modelling of atmospheric stability conditions, number of simulations and so on. Please clarify.
    ⇒ *Agreed, changed to: "Despite the lower resolution when compared to microscale CFD simulations, they can assess a large variety of different inflow conditions in a very computationally efficient manner (Machefaux et al., 2015)."*

7. L72 "… benefits of grid-less wake models", this might be a new term to the reader, please can you explain a little more about grid-less models, and their benefits.
    ⇒*Agree, sentence changed to: "In this study, we propose a grid-less approach that unlocks the computational benefits of using analytical engineering wake models, that do not necessitate of a grid to be resolved."*

8. L83 About "investigate", perhaps a better word is assess or evaluate. Investigate is a bit too general.
    ⇒ *Thanks, changed to "evaluate"*

9. L120 About "normed", please can this process be clarified. I think it refers to how many degrees of turning there is per 100 km.
    ⇒ *Yes, it is. Normed was misleading here. We reformulated the sentence to: "In order to account for the distances to the reference location, the wind direction changes per 100 km were evaluated using the great circle distance between the $P_j$ and $P_{REF}$."*

10. Fig 1: Nice map, I suggest the different scales and the lines are given different colours to make them stand out a bit more.
    ⇒ *Thanks, distances are given in red now.*

11. L126 Heading, consider adding "*synoptic* pressure systems"
    ⇒ *Changed accordingly. Thanks.*

12. L129 "drop in *mean sea level* pressure" or "drop in *surface* pressure ⇒ *Changed to "drop in surface pressure".*
    *Thanks.*

13. L155 Eq 4 and 5. I think it would be nice with more guidance to the reader about this coordinate system part. For example a diagram or two, showing how the coordinate system looks in relation the streamlines and perhaps a visualisation of the wake getting curvature in some way.
    ⇒ *We added a new figure [Figure 3 now] to Section 2.2, showing the relation of the piecewise-linear streamline and the local coordinate systems. Also, a visualisation for the example of the Jensen wake model in heterogeneous background flow with wind rotation has been added. Hopefully this clarifies Eqs. (4) and (5). The text in Section 2.2 has been extended and now describes the method in more detail.*

14. L185 What is "REWS"? It is not explained anywhere. Please clarify.
    ⇒ *REWS stands for "rotor equivalent wind speed" that was introduced in line 174 (old manuscript) and commonly used in wind energy science. It is the average velocity across the rotor that can be used to derive thrust and power from the manufacturer curves.*

15. L196 "emblematic" is an odd word choice, I think "indicating" would be better.
    ⇒ *Thanks, changed to: "Additionally, single situations spotted through SAR imagery to indicate the wake turning phenomenon were compared to* flappy *simulations spanning the whole German Bight."*

16. L218 It is a bit unclear here what was chosen for the model coefficients. In some places low $k_{TI}$ and $k_b$ is used to make wakes deeper and prolonger, those also more realistic values are used too.
    ⇒ *To make it clearer, the paragraph has been rephrased in the revised version to:*
    *"For the results of Sect. 5.2.1, where the focus it on showcasing just the computed wake trajectory, the wake model parameters were adjusted to minimize wake recovery and to achieve slowly decaying wakes. The near wake length was set to zero, $k_{TI} = 0.05$ and $k_b = 0.0$. Such a set of parameters exacerbates unrealistically the power loss due to wake effects within the farm, and it is not advisable for computing a meaningful energy yield of the two clusters. Therefore, they are not suited for a meaningful quantitative analysis. The parameter used for the quantitative evaluation in sect. 5.2.3 were calibrated using an optimization procedure involving the supervisory control and data acquisition (SCADA) data of one of the wind farms in the wind farm clusters considered. The procedure, here not presented for brevity, determined the coefficients $k_{TI} = 0.23$, $k_b = 0.003$, $\alpha = 1.4$, and $\beta = 0.077$ to guarantee the best agreement between the engineering model and the real production data."*

17. L223 About the calibration of coefficients, did that use the BLM SWM set-up? Would it make a difference which was used?
    ⇒ *The calibration procedure was done with the BLM model. It involved a comparison to actual SCADA data in a single farm. The inflow conditions for the engineering model were defined according to the SCADA as a timeseries of uniform values. With this input the BLM and the SVM model produce the very same results, as no direction change is present in the flow field.*

18. L237 and Eq 8 "REWS" is used again. What is it?
    ⇒ *see above - Comment 14 - Line 185 (old manuscript).*

19. Fig 3: Small thing. In the main text and Table 1 <14 m/s is referred to, but in the figure the binning uses 15 m/s.
    ⇒ *True. This might be somehow inconsistent, but we suggest staying with this. The figure uses a constant binning of 2.5 m/s and in Table 1 we use the actual values that are based on typical technical specifications of wind turbines in the German Bight. We refer to the table a bit earlier in the text now and changed the order of figure and table.*

20. L290 and thereabouts: can the authors say anything about the asymmetry of the turning characteristics. I think this is pretty interesting.
⇒ *The asymmetry is in particular caused by coastal effects. During onshore/offshore oriented winds, the coastal transition leads to a drop in the surface roughness and a new force balance between pressure, friction and Coriolis force and thus a turning of the wind along the coastline (see e.g. Emeis, 2018). We reformulated a paragraph of this section: "The asymmetry in the wind direction changes is mainly caused by the coastal transition and can thus in particular be observed at the eastern and southern coastlines of the German Bight and for moderate to high wind speeds (see e.g. Fig. 5). There, comparably large direction changes can reach up to some tens of kilometers offshore."*

21. L299 Again the "normed" term again, please correct. Also in caption of Fig. 4.
⇒ *Changed, also in the caption and in the (sub-)figures.*

22. Fig 4: I like these graphs. Not seen anything like this before.
⇒ *Thanks!*

23. Fig 5: The colour scale has "KDE" please explain what this means, and the caption spells it out, but how does the reader understand it. Does it have a unit?
⇒ *KDE stands for (Gaussian) kernel density estimation with the unit percentage and indicates the density of the points in the area. We modified the caption to: "The color shows a Gaussian Kernel density estimation (KDE)."*

24. L332 "exemplary" is not the correct word I'd say. "Example" is better. ⇒ *"Exemplary showcase" changed to "particular example"!*

25. L393 From what is shown in the article, I think it is difficult to claim the extent is significantly greater in models. The figure 8 has different scales, and not even the plotted quantities are the same. For SAR, it's the NRCS and not the wind speed.
⇒ *That's fair to point out. We meant that the wakes were vanishing earlier in the SAR data than in the models. However, the whole paragraph has been restructured, and such claim is not present anymore: "Contrasting the result presented above, in other situations analyzed the agreement between SAR and both modeling strategies was poor (not shown). However, this mismatch originates from a disagreement between SAR and the mesoscale input data. Therefore, in the context of engineering modelling, these results do not diminish the attractiveness of the SWM against the BLM, as also the second would suffer from the same issue."*

26. L425 "fade away" replace with "reduce".
⇒ *Done, thanks!*

27. L440 Why is it the "uncertainty" that is reduced here? Isn't it just the difference in the two estimates? Why is that uncertainty?
⇒ *It is just the difference between models. However, as the SWM should capture a physical phenomenon not accounted for in the baseline, in case it was validated, it should reduce the engineering model uncertainty. However, since this is just speculation, the paragraph has been rephrased: "In conclusion, in our result, the proposed model offers a small but non-negligible difference when estimating the interaction between neighboring wind farm clusters, as N6 and N8. Despite the new model is not yet validated with production data, we speculate that such a difference could represent an actual improvement over the baseline in terms of model accuracy."*

28. L447 and onwards: This is a good point for discussion. I think the authors could consider some scale of interest here. For example, 100 km takes 167 minutes (nearly 3 hours) at 10 m/s. While the update time from NEWA is 30 minutes. At 10 m/s the advection would be 18 km. In the future, it would be nice if there was validation against SCADA data so the stationarity assumption of the streamlines could be tested over different time and length scales. One hypothesis could be that for intra-cluster scales it may be ok, while for German Bight scales the assumption is not so suitable. In this case, the wake effects may be much smaller anyway.
⇒ *Thanks for this comment. A future work involving SCADA is already planned for trying to verify the implication of the stationary assumption. Similarly, could be also achieved by comparing to WRF with or without the farm parametrization, and with a finer temporal resolution. We modified the outlook at the very end of the manuscript and added a comment that the stationarity assumption should be validated with production data as well:*

*"Future development, partly already initiated, encompasses the attempt of validating the model with a combination of lidar measurements and wind farm production data, and the comparison to other large-scale cluster wake modelling approaches such as mesoscale wake modeling (Fischereit et al., 2022a). The study by Fischereit et al. (2022b) provides an attempt of comparing different wake models of different complexity although on a smaller scale than the German Bight. This validation should then also focus on the validity of the assumption of stationarity of the streamlines."*

29. L486 About "far greater fidelity", I think this was the case in the examples shown with large amounts of streamline turning. One could add the comment that the BLM showed wakes crossing each other, and this was, I'd argue, a non-physical situation.
    ⇒ *"Far greater fidelity" has been substitute with "in a way more consistent to physics", in order to make clear what pointed out in your comment.*

30. L488 I'd say the "accuracy" can only truly be assessed with comparison against SCADA data. The conclusion about the accuracy depending on the accuracy of the underlying NEWA data, I say is actually only at this stage a reasonable hypothesis, because it is not actually tested in the article, as far as I can see.
    ⇒ *From the comparison to SAR data, we found situations where the flow field between model and measurements was substantially different. In such occasions, the computed production of the farm would be in disagreement with the SCADA data. As we root the cause of these differences mostly in the mesoscale input field, we think it is important to underline how the accuracy of the input flow fields strongly influence the outcome of the simulations with the engineering models. We reformulated this paragraph to:*
    *"A qualitative comparison of the simulated flow fields around the wind farm clusters in the German Bight with SAR data shows that the new model can represent the flow in a way more consistent to physics than the common baseline approach. We conclude that the ability of the model to represent large-scale direction changes depends greatly on the accuracy of the underlying data (NEWA data in the discussed cases) in representing those. However, in no occasion the BLM provided a better representation of the wake propagation than the SWM."*

31. L495 About "reduction in uncertainty", again this reasonable hypothesis, but not tested again SCADA data. The article describes differences in the two approaches, however it is not clear to me that one is absolutely shown to better than the other. This would require validation against SCADA data.
    ⇒ *We agree with this point of view. We rephrased the line in question, and other parts of the paper, to make clear that we only speculate that the SWM is better than the BLM (see last sentence of comment above). In case this was validated, then we suggest the difference between the models to be a measure of accuracy gain.*

**Further Changes:**

1. The results in this paper were obtained using the Fraunhofer IWES in-house code *flappy*, which has recently been released as open-source software under the new name *FOXES - Farm Optimization and eXtended yield Evaluation Software*. This has been mentioned in the manuscript under Section 3.1 and *Code and data availability*.

**Reply to Reviewer 2 by Anna von Brandis et al**

This paper covers an important topic, namely how spatial variability in wind direction affect the evolution of wind farm cluster wakes and how this can be implemented in engineering wake modelling tools. I do however have some substantial concerns that need to be address before I can recommend the paper for acceptation in wind energy science. The improvements necessary are detailed below:

⇒ *We thank the reviewer for the careful evaluation of our manuscript and the comments to the work presented. We address all the comments of the reviewer separately below.*

**Major comments:**

1. "spatial variability in wind direction" covers better the content of the paper than "mesoscale wind direction changes" and should in my opinion be used throughout the paper. First of all, the systems that are discussed here are often larger than what is traditionally described as mesoscale systems. Moreover, the paper focuses on variability in space and not in time (streamline instead of lagrangian approach).

   ⇒ *We partly disagree with this comment. We think that in the main part of the manuscript the methodology, underlying datasets and figures clearly enough describe and demonstrate the associated scales of our investigations. We agree however to change the title to "An investigation of spatial wind direction variability and its consideration in engineering models".*

2. The qualitative comparison with synthetic aperture radar (SAR) is the center of the paper, but the setup of the model experiments and comparison with data should be improved. To produce this figure, the wake model parameters were adjusted to minimise wake recovered and to achieve slowly decaying wakes. However, the authors argue that this does not yield a goods agreement with real production data and they propose other parameters for realistic simulations. It would therefore be much fairer to produce the maps and the comparison with SAR for realistic coefficients of the wake models. Moreover, the current comparison is not straightforeward to interpret since SAR data in figure 8 are displayed as the backscatter value of the normalised radar cross-section. It would be much better to use here the geophysical quantity wind speed which also improves the ability to compare the observed versus the modelled far wakes. ⇒ *Although the comparison with SAR is important for our conclusions, we do not see it as the focal part of the paper. The main purpose of Fig. 9 [initial version: Fig. 8] is to provide evidence of the wake turning due to mesoscale wind distribution through the SAR data and to demonstrate that such turning can be represented reasonably well only with the SWM algorithm. Hence, the focus is on the trajectory that the cluster wakes have rather than their extent. For this purpose, we believe that the best way to let the reader focus on the wake trajectory predicted by the models is to avoid vanishing wakes. Thus, we decided to tailor the wake model to have a negligible wake recovery. Furthermore, we wanted to avoid discussing cluster wake extent and velocity deficit. Therefore, we don't find necessary to convert the SAR data into wind speed fields, as this would also add further uncertainty. We have however reformulated the paragraph about the definition of the wake model coefficient to describe our intention in more detail:*
   *"For the results of Sect. 5.2.1, where the focus it on showcasing just the computed wake trajectory, the wake model parameters were adjusted to minimize wake recovery and to achieve slowly decaying wakes. The near wake length was set to zero, $k_{TI} = 0.05$ and $k_b = 0.0$. Such a set of parameters exacerbates unrealistically the power loss due to wake effects within the farm, and it is not advisable for computing a meaningful energy yield of the two clusters. Therefore, they are not suited for a meaningful quantitative analysis. The parameter used for the quantitative evaluation in sect. 5.2.3 were calibrated using an optimization procedure involving the supervisory control and data acquisition (SCADA) data of one of the wind farms in the wind farm clusters considered. The procedure, here not presented for brevity, determined the coefficients $k_{TI} = 0.23$, $k_b = 0.003$, $\alpha = 1.4$, and $\beta = 0.077$ to guarantee the best agreement between the engineering model and the real production data."*

3. I am not convinced with the use of the term "reduction in uncertainty" in the paper. In my opinion what the authors propose here is rather an attempt to include a process, namely the curved propagation of cluster wakes as a result of spatial variability in wind direction and thereby improve existing wake models. Even though this improvements cannot be quantitatively demonstrated in the current framework. For example, in line 440 where you state that applying the proposed model reduces uncertainty by up to 0.7 % when estimating the interaction between neighboring wind farm clusters. This is just referring to a sensitivity where curvature of the path is included versus a situation where this is not included and is not a metric for reduced uncertainty. Please go through the entire paper to critically reflect on the use of the term uncertainty and reduction of uncertainty.

   ⇒ *Thank you for this critique, the term was indeed used with too much liberty in the context of the paper. We are convinced that considering wake turning is fundamental for a correct estimation of how neighboring wind farms hamper each other performances. Therefore, we were eager to quantify how including this phenomenon would have affected the*

*engineering model results. We now made very clear that only in the case the SWM model is validated and demonstrated better than the BLM, the differences across each other are a step forward in the engineering model accuracy. We modified several parts of the manuscript to reflect this in prticular in Section 5 and on the Conclusions.*

4. In Section 5.1.1, I do not understand the logic of taking the mean of the medians. If I understand it correctly, you first take for every time frame the difference in wind direction compared to FINO. Then you take for each pixel the yearly median of this difference. Subsequently, you average over a 30 year period. Why making it so complex and not just taking the median of all the timeframes for each pixel (or the mean if you prefer)?
   ⇒ *We didn't take the mean as the wind direction changes can be varying quite significantly into and the median provides a better statistical representation of what we want to show. In particular, in Figure 5 [formerly Figure 4], we want to show how the upper 25 % and lower 25 % that contribute in a significant share to the long-term conditions. The median (aka 50th percentile) is the corresponding value to 25th and 75th percentile. The mean then reflects averaging over a thirty year climatology.*

5. In the same section, I do not understand why your value does not become zero when looking at the FINO grid point for 25th percentile and the 75th percentile. Per definition, the direction difference with reference to FINO should always be zero at FINO. Moreover, there is little discussion anyway on the 25 and 75th percentile, so it is perhaps better to remove this.
   ⇒ *You are right, we described this in the text but not in the Figure caption that the data are not based on a single point but an average over a larger area (gray area in Figure 2). We added this information to the caption as well. In addition, we added another sentence to interpret the results of Table 2: "Table 2 also reflects that while the mean wind direction change can be quite low, a large share of situations in particular for wind speeds in the partial load range ($< 10.0\,ms^{-1}$) exist with significant wind direction changes."*

6. In the same section, I am not sure if it does make sense to take the median (or the mean) of a direction (which is a circular metric). The average wind direction of -179 deg and +179 deg is not 0 deg. Am I missing something on how you exactly did the data processing (and then you can perhaps explain it better). Of course if the difference in the direction is in between -90deg and +90deg it is okay, but I would expect that larger differences can occur over a distance of 200 km or so? Even if you scale it back to 100 km there might be similar issues. Please check.
   ⇒ *Of course, we did account for the circular properties of the wind direction variable. We didn't calculate the average wind direction, but the average/median of the CHANGE of the wind direction. The average wind direction of e.g. of 359 degrees and 1 degrees would be 180 arithmetically but in reality a wind from North (i.e. 360 degrees). In our case, we reflected this by subtracting/adding 360 degrees in case a direction change is larger/smaller than +/-180 degree. To give an example: When we calculate the wind direction difference of a wind from 359 and 1 the resulting difference in our case is 2 degrees. Larger differences approaching +/- 180 degrees over 200 km do occur, but very rarely. This could be caused by e.g. winds at a front / strong convection. This is by the way reflected in the scatter plot in Fig. 6 [formerly Fig. 5], where some cases of wind direction changes of > +/− 100 degree exist. We added one sentence to the Method section (2.1.1): "In all cases where wind direction differences were calculated, we accounted for the circular nature of the wind direction variable."*

7. Figure 7 and 8: add the positions of the wind farms to this figure. Moreover, make a map using a lat/lon grid similarly to the other figures that you created. Make Figure 8 so that you can directly compare the model with SAR by using the same map and also by plotting wind speed in both the model and SAR.
   ⇒ *We don't understand why the reviewer suggests adding the wind farms to these figures. The individual wakes of the turbines and their origin are visible in Figures 7 and 8 [revised version: Figures 8 and 9]. In addition, the names of the wind farm clusters are given in the figures. The actual positions of the individual turbines are given in Figures 1 and 2. If we add those to Figures 7 and 8 again, we risk showing too much information and distract the reader from the wakes that we want to show in these figures.*

8. Figure 9: you plot the standard deviation of the relative difference. I am a bit puzzled why this is an interesting metric. Why not simply plot the relative absolute difference, which is a much easier metric to intuitively interpret. The standard deviation will also never be a negative number, so what you plot in the figure cannot be a standard deviation, since negative numbers appear in the figure.
   ⇒ *The caption of the figure has been updated to better clarify that, in the heatmap, only the color of each bin represents the standard deviation, for which there is indeed no negative value. The numbers annotated within each bin are instead the relative absolute difference (or mean of the relative difference across all the observations in a bin). We think that it is important to show the standard deviation because it suggests that the differences between the two considered models*

*are very high on a single situation basis.*

9. Overall after reading your paper, although I'm not an expert on wake modelling, your results seem to imply that turning of the wakes in relation to spatial variability in wind direction is relevant, but that the current engineer models of the wind farm wakes are perhaps not appropriate yet to firmly demonstrate this. Your conclusion on line 486 is that "it is concluded that the new model can represent the flow with the fact greater fidelity than becoming baseline approach." This statement is much too strong given the limitations of the analysis currently done and discussed above.
⇒ *The statement has been changed to "in a way more consistent to physics". Such claim refers to current Fig. 8, where we demonstrate the BLM to predict two wakes crossing each other, a rather non-physical situation. Concerning the conclusion that engineering models are not suitable to demonstrate the effect of wake turning, it is important to consider that accounting for this phenomena can easily either produce benefits or malus to the power production of the farm considered when we sum over all the differences to compute yield they tend to balance out. However, it is reasonable that shortening the period of yield evaluation or changing the location and the number of wind farms could provide a different result.*

10. Also the line 488 doesn't clearly follow from the results where you state that "the new model should consistently outperform the baseline approach that neglects wake turning." Well perhaps in principle it should (because in reality the wakes move with the flow) but the paper does not firmly demonstrate that.
⇒ *Thanks. We changed the sentence to: "in no occasion the BLM provided a better representation of the wake propagation than the SWM."*

11. A related question: are there better ways to model the wake of a wind farm than adding up the wakes of the individual turbines? In figure 7, the wakes of the individual turbines remain visible over a long distance and I wonder whether there is observational evidence for this: Individual streaks remain visible in the model as far as 50 km downstream. Is this realistic? I therefore would like to see a brief discussion on this together with a discussion on how wakes of farms are implemented in current engineering models, how this model relates to other approaches and whether the approach presented here is accurate enough to study wind farm wakes and interaction between wind farms.
⇒ *In Fig. 7 we wanted to demonstrate the full path followed by the wakes within the domain according to the different models. For this reason, we tweak the wake model to reduce wake recovery to a minimum. With more pertinent wake model coefficient, such as the one used in the quantitative comparison of SWM and BLM the single wake streaks will decay much faster into a wind farm wake. So far, despite the superposition of single turbine wakes presents many flaws, it is still the computationally cheapest way to compute wind farm yield with acceptable level of confidence. The comparison of different large-scale wake modelling approaches of different complexity is a very recent topic in wind energy science. We added reference to a very recent study that compares mesoscale and engineering wake models in the outlook "Future development, partly already initiated, encompasses the attempt of validating the model with a combination of lidar measurements and wind farm production data, and the comparison to other large-scale cluster wake modelling approaches such as mesoscale wake modeling (Fischereit et al., 2022a). The study by Fischereit et al. (2022b) provides an attempt of comparing different wake models of different complexity although on a smaller scale than the German Bight."*

**Minor comments:**

1. It would be good to add a short description how the wake expansion is related to the atmospheric conditions. From the equation 6, it is clear that the wake deficit is in turn a function of ambient turbulent intensity, which is a function of meteorological conditions. A paragraph explaining how this relation is exactly formulated would add strong value to the paper. Later in the paper, this can be used to describe how the wake expansion relates to meteorological conditions in the cases that are presented (low pressure system).
⇒ *Thanks for this comment, we added a couple of sentences to section 3.2 of the revised manuscript to reflect this point: "The wake expansion coefficient, $k*$, determines the length of the wakes, and it should be connected to the meteorological conditions of the atmosphere, i.e. atmospheric stratification. However, it still remains unclear in the framework of engineering modelling how to express such a dependency. In this work, we adopted a linear relation on the turbulence intensity that should at least partially capture how different atmospheric stratification affects the wake recovery."*

2. Section 2.2 on the implementation of direction changes into engineering models should be improved. It is unclear what x exactly means. Moreover, r(t) is the path as a function of time whereas s is distance along a path. The distinction should be better explained . More details should be given on the determination of the coefficients of the engineering model for the realistic situation. This is not a detail of the paper, because the extent of the wake and consequently the

turning of the wake, is very much determined by these parameters therefore the potential impact of spatial variability in wind direction is very much dependent on these parameters.

⇒ *We added a new figre (now: Figure 3) to Section 2.2, showing the relation of the piecewise-linear streamline and the local coordinate systems. Also, a visualization for the example of the Jensen wake model in heterogeneous background flow with wind rotation has been added. Hopefully this clarifies Eqs. (4) and (5). In steady-state flow, any point $\mathbf{p}$ along the streamline curve can either be described as a function of the travelling time $t$, or, equivalently, as a function of the path length $s$, counting from the wake-causing rotor. The text in Section 2.2 has been extended and now describes the method in more detail.*

3. In Section 5.1.1 it is shown and discussed that wind direction are large for very high wind bins. Can you analyse the cause of this? Are these situations where the location is close to the central point of a deep low pressure system with strong wind speed but also strong curvature?

   ⇒ *Thanks for this comment. We are not showing particular situations here, but it is very likely that these large changes in wind direction are associated with e.g. the passage of cold fronts of low pressure systems that typically are associated of with a strong change in wind direction from southerly to more westerly or even northwesterly winds. We added one sentence to discuss this:*
   *"Very high wind speeds, despite less frequent, come along with larger positive direction changes. One explanation for this could be the passage of cold fronts of low pressure systems that typically are associated of with a strong change in wind direction from southerly to more westerly or even northwesterly winds."*

4. Please look again figure 5 on what you plot on the x-axis. You take the logarithm of a variable and then you plot this on a logarithmic scale.

   ⇒ *Thanks for pointing this out. The log was removed from the axis label.*

**Further Changes:**

1. The results in this paper were obtained using the Fraunhofer IWES in-house code *flappy*, which has recently been released as open-source software under the new name *FOXES - Farm Optimization and eXtended yield Evaluation Software*. This has been mentioned in the manuscript under Section 3.1 and *Code and data availability*.

---

## Author Response (AR2)

**Second comments of Reviewer 1 for Anna von Brandis et al**

1. L120 [original line number] About "normed", please can this process be clarified. I think it refers to how many degrees of turning there is per 100 km. The comment is some addressed. Perhaps better would be: "..the wind direction changes *expressed as degrees* per 100 km were evaluated... "

2. L299 [original line number] Again the "normed" term again, please correct. Also in caption of Fig. 4.
   ⇒ *The word "normed" has been fully replaced by "per 100Km"*

**Second Comments of Reviewer 2 for Anna von Brandis et al**

1. There is just one point remaining, namely point 4: My point was that now the authors take the yearly median of the change and then average this over 30 years. I suggest to take the median of all timeframes in the 30 years, if this would still be possible.

**Answer of the Authors**
⇒ *We kindly thank both the reviewers for insisting on improving section 5.1.1, dealing with the wind direction changes "normed over 100Km" or "per 100Km" (figure 5 and Table 2). After an additional round of internal discussion, we established that the display of such a quantity could cause confusion, mostly on the procedure through which it is derived. We finally decided to entirely remove the normalization of wind direction changes from the manuscript. All the wind direction changes now showed are the actual wind direction differences with respect to the reference location. For this reason, the current figure 5 and Table 2 have been updated and their content changed. However, the message they convey remains identical. The same figure and table have been also updated considering the comment from Reviewer 2. Therefore, median and quartiles are computed from the whole 30 years dataset directly and no longer as the mean of the yearly values.*